# Remotely sensed and surface measurement derived mass-conserving inversion of daily NOx emissions and inferred combustion technologies in energy rich Northern China

Xiaolu Li[1,2], Jason Blake Cohen[2]*, Kai Qin[2]*, Hong Geng[1], Xiaohui Wu[3], Liling Wu[4], Chengli Yang[3],
Rui Zhang[5], Liqin Zhang[6]

[1]Institute of Environmental Science, Shanxi University, Taiyuan, 030006, China
[2]School of Environment and Spatial Informatics, China University of Mining and Technology, Xuzhou, 221116, China
[3]Shanxi Dadi Ecology and Environment Technology Research Institute Ltd., Taiyuan, 030000, China
[4]School of Environment, Tsinghua University, Beijing, 10084, China
[5]Shanxi Institute of Ecology and Environment Planning and Technology, Taiyuan, 030002, China
[6]Shanxi Institute of Ecology and Environment Monitoring and Emergency Response Center, Taiyuan, 030027, China

*Correspondence to*: Jason Blake Cohen (jasonbc@alum.mit.edu) and Kai Qin (qinkai@cumt.edu.cn)

**Abstract.** This work presents a new model free inversion estimation framework using daily TROPOMI $NO_2$ columns and observed fluxes from the continuous emissions monitoring system (CEMS) to quantify three years of daily-scale emissions of $NO_x$ at $0.05° \times 0.05°$ over Shanxi Province, a major world-wide energy producing and consuming region. The $NO_x$ emissions, day-to-day variability, and uncertainty on a climatological basis are computed to be 1.86, 1.03, and 1.05 Tg per year respectively. The highest emissions are concentrated in the lower Fen River valley, which accounts for 25% of the area, 53% of the $NO_x$ emissions, and 72% of CEMS sources. Two major forcing factors ($10^{th}$ to $90^{th}$ percentile) are horizontal transport distance per day (63-508 km) and lifetime of $NO_x$ (7.1-18.1 h). Both of these values are consistent with $NO_x$ emissions to both the surface layer and the free troposphere. The third forcing factor, the ratio of $NO_x/NO_2$, on a pixel-by-pixel basis is demonstrated to correlate with the combustion temperature and energy efficiency of large energy consuming sources. Specifically, thermal power plants, cement, and iron and steel companies have a relatively high $NO_x/NO_2$ ratio, while coking, industrial boilers, and aluminium oxide factories show a relatively lower ratio. Variance maximization is applied to daily TROPOMI $NO_2$ columns, which facilitates identification of three orthogonal and statistically significant modes of variability, and successfully attributes them both spatially and temporally to (a) this work's computed emissions, (b) remotely sensed TROPOMI UVAI, and (c) computed transport based on TROPOMI $NO_2$.

**Graphical abstract.**

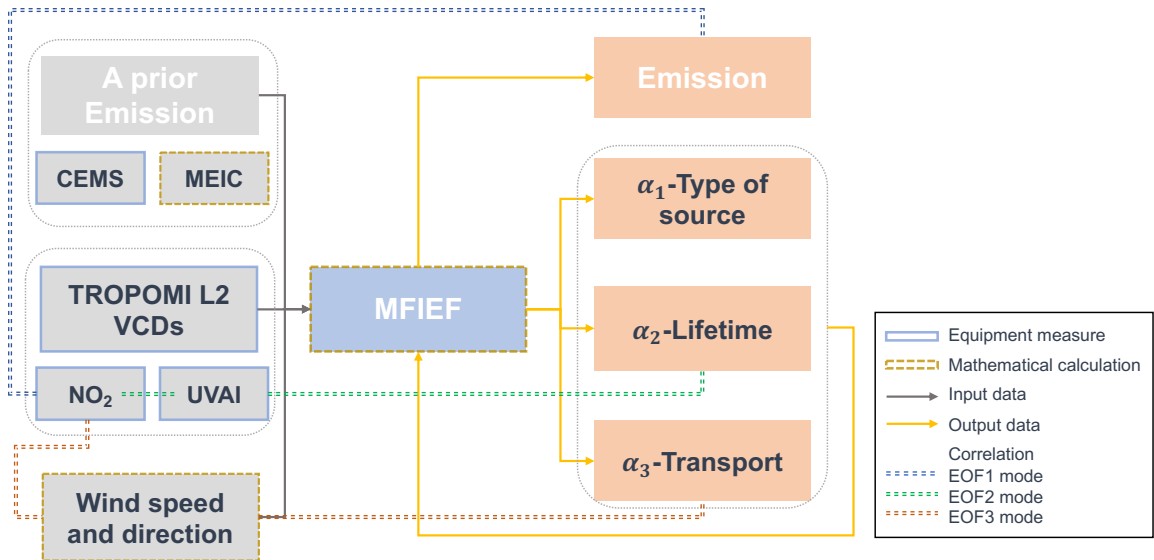

35

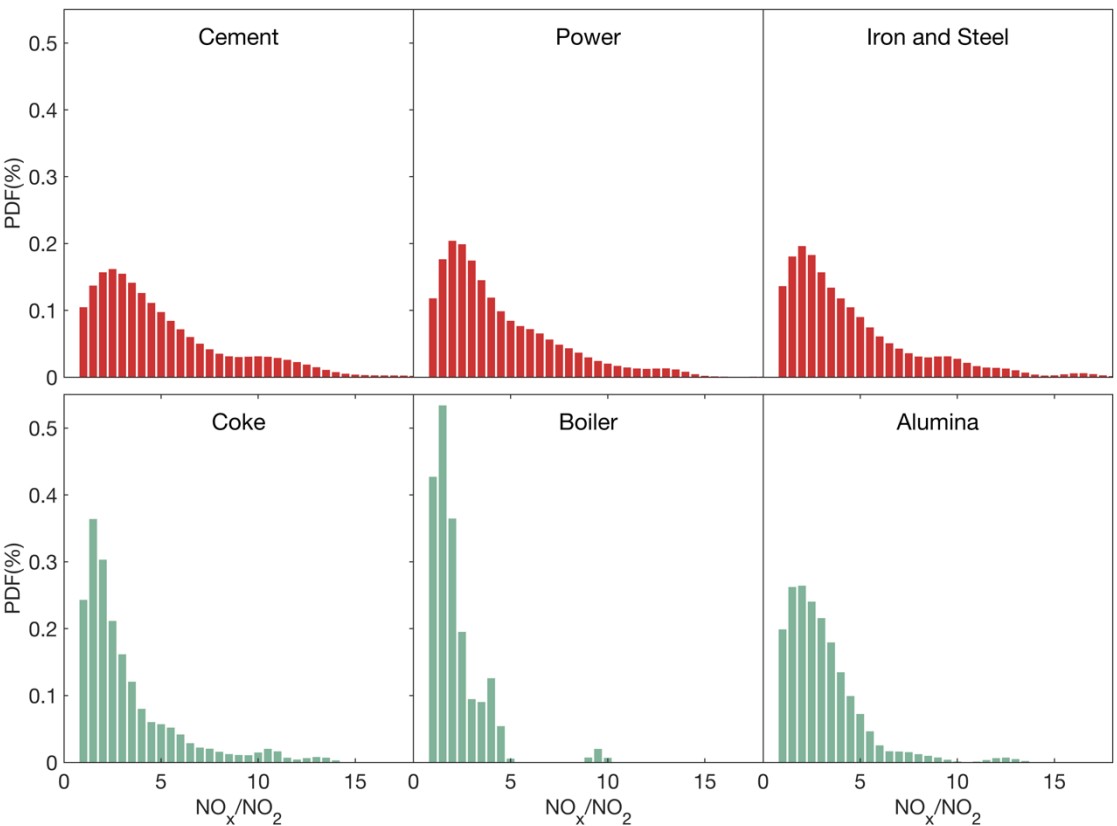

# 1 Introduction

Economic growth contributes to the emissions of air pollution, leading to serious environmental and health consequences. To address and alleviate the air quality problems, the Chinese government has implemented continuous air pollution controls, with the aim of producing higher-quality development. Two recent examples are the Air Pollution Prevention and Control Action Plan from 2013 to 2017 and the Three-Year Action Plan for Winning the Battle in Defense of Blue Sky from 2018 to 2020 (Zhang et al., 2019; Geng et al., 2019; Jiang et al., 2021; Wang et al., 2020b; Li et al., 2022a; Wei et al., 2023), which have led to a significant reduction in annual average concentration of particulate matter (PM), sulfur dioxide ($SO_2$) and carbon monoxide (CO) in Shanxi Province. Shanxi is selected for this study, with geographical location and topography showed in Fig. 1, as it is a highly energy rich location that produces nearly 30% of China's coal, as well as having substantial industry that consumes a significant amount of coal for local energy production, steel, cement, coke, and aluminum production, and export, among other economic activities (Li et al., 2022). Moreover, there have also been minor increases in the observed annual average concentrations of both ozone ($O_3$) and nitrogen dioxide ($NO_2$) in Shanxi between 2015 and 2020 (DEESP, 2015; DEESP, 2020). Furthermore, due to its relatively dry climate, high elevation, and mountainous geography, it has complex underlying natural factors that impact its atmospheric environment.

The sum of $NO_2$ and nitric oxide (NO) is frequently grouped as nitrogen oxides (herein termed $NO_x$), which is an important trace gas impacting of the earth's atmosphere because it is a strong marker of anthropogenic combustion-related pollution, a precursor to ozone (Jacob et al., 1993), secondary aerosol (Rollins et al., 2012) and acid rain (Singh and Agrawal, 2007). In order to gain a better understanding of $NO_x$ and its impacts, precise and quantitative emission inventories are crucial information for policy makers, air quality modelers, climate change modelers, and those who conduct pollution weather response interactions, among others (Hoesly et al., 2018; Crippa et al., 2018; Li et al., 2017a; Xing et al., 2013; Mcdonald et al., 2013). However, it is challenging to quantify emissions in rapidly developing and changing areas accurately as there are a variety of contributing sources, a complex underlying mixture of combustion technologies, industrial restructuring, changing population dynamics, and ongoing atmospheric environmental management, all of which may lead to substantial changes in pollution sources.

Presently, most emission inventories are compiled from statistics representing emitting activities and associated typical emission factors, herein called "bottom-up" approaches (Ohara et al., 2007; Zheng et al., 2018; Li et al., 2017b). Bottom-up methods provide emissions data at finer scales. However, higher resolution emission inventories using bottom-up methods require rich, detailed, and extremely precise records of energy use, facility locations, and other socioeconomic datasets from multiple regional and temporal scales (Cai et al., 2018), which frequently have a considerable amount of uncertainty (Bond et al., 2007). With the ever-increasing tightening of environmental management, emission factors have changed significantly and will continue to do so in the future. On-site surveys for bottom-up methods are time consuming and resource demanding. When performing emission factor determination in laboratory, it is important to note that the differences between small field studies and controlled laboratory combustion experiments and real-world examples also are quite significant, with super-emitters

known to create large differences when using insufficiently large datasets (Zavala et al., 2006) and missing large sources leading to significant error (Wang et al., 2021). Due to the low temporal resolution and time-lag associated with many of these datasets being available, bottom-up inventories are not easy to keep up with rapid changes in industrial, economic, and pandemic, and therefore are not very good at tracking atmospheric emissions under actual existing environmental conditions, limiting their use (Mijling and Van Der A, 2012).

To overcome the disadvantages identified above, while simultaneously improving the spatial and temporal resolution of emission inventories, attempts at top-down emission inventories using remote sensed dataset have been made by the community. Some of these attempts have focused on applications to long-lived gasses ($CH_4$, CFCs, and $N_2O$), since their chemical decay is slow compared with their transport processes, allowing a simpler set of approximations to perform the inversion (Chen and Prinn, 2006; Tu et al., 2022b; Liu et al., 2021). Satellite observations have also been widely used to quantify short-lived species such as $NO_x$ emissions (Martin, 2003; Beirle et al., 2019; Goldberg et al., 2019; Qu et al., 2019) by providing up-to-date and continuous time series of $NO_2$ columns in different regions. Gaussian plumes have been used for a long time (Green et al., 1980), with more advanced but similar approximations including the exponentially modified gaussian model to quantify the $NO_x$ emissions of isolated megacity sources (Beirle et al., 2011) and the probabilistic collocation method to train the emissions flux enhancement of megacities as a function of their size and shape (Cohen and Prinn, 2011). Some methods used a partial estimation of the mass balance approaches, including Beirle et al. (2019) and Kong et al. (2019). Another category based on atmospheric chemical transport models, climate models, and/or data assimilation, such as 3D-Var, 4D-Var, and Kalman filters work very well but are susceptible to underlying model and scientific uncertainty, as well as being extremely computationally intensive (Cohen and Wang, 2014; Zhang et al., 2021). The selection of a priori inventories is crucial when using the methods above, since it has been demonstrated that missing sources can frequently not be compensated for by merely scaling or increasing other existing sources if their spatial and temporal distributions are not matched correctly (Cohen, 2014). In recent years, continuous emissions monitoring systems (CEMS) have been introduced in China, the USA, and other countries and areas, as a means to detect in an integrated manner at the emissions effluent source, on an hourly and/or daily time scale, and contains well-known aspects of quality control and assurance. This platform provides reliable technical information to fundamentally quantify local emissions on a stack-by-stack basis, in order to improve both the temporal resolution and magnitude of emission inventories (Tang et al., 2019; Gu et al., 2022; Chen et al., 2019; Lange et al., 2022).

This study takes advantage of the respective strengths of top-down and bottom-up emissions estimation by applying a new, fast, first-order approximation of physical, chemical, and thermodynamics controlling the distribution of $NO_x$ in-situ, and constrains these approximations using daily measurements of remotely sensed $NO_2$ from the Tropospheric Monitoring Instrument (TROPOMI) together with a mass conserving inversion to estimate the daily $NO_x$ emissions on a mesoscale grid at ($0.05° \times 0.05°$) from January 2019 through December 2021. This approach is partly originated from similar box-modeling ideas in previous studies (Rigby et al., 2008; Beirle et al., 2019; Kong et al., 2019), which themselves are based on previous theory underlying the development of mass-conserving box models (Seigneur et al., 1986). In this specific work, the mass of emissions is connected with the in-situ observed column loadings through application of the following factors: the temporal

rate of change in column loadings, first order chemical loss of $NO_x$, gradient transport of $NO_x$, and gradient transport of atmospheric airmass. The coefficients weighting these terms are flexibly fitted, allowing a wider range of possible driving forces and solutions to be considered, while still requiring that these parameters are consistent with observations (Rollins et al., 2012; Karl et al., 2023). The fitted relationship is formed without the use of complex models, can be run on a normal desktop computer, and the end product can be flexibly modified by the user for their own various applications. The a prior emissions used in this work come from daily CEMS observations at power plants and other large sources, as well as the Multi-resolution Emission Inventory for China (MEIC) over Shanxi province. This unique perspective is capable of using measured emissions, fitting variable parameters that were fixed in previous approaches, and inverting emissions as a function of their month-to-month constrained driving forces, under different but realistic environmental conditions. This method has been used in different situation such as over different months, over multi-year changes in the environment, under different actinic flux and atmospheric oxidation conditions, under complex meteorological domains, and over sources which are both thermodynamically stable as well as unstable. That permits this study to explore the full range of variations. Additionally, this approach allows for results with both a robust error quantification and emissions that compare well with both the mean and the measured spatial and temporal variation in the underlying remotely sensed $NO_2$ columns.

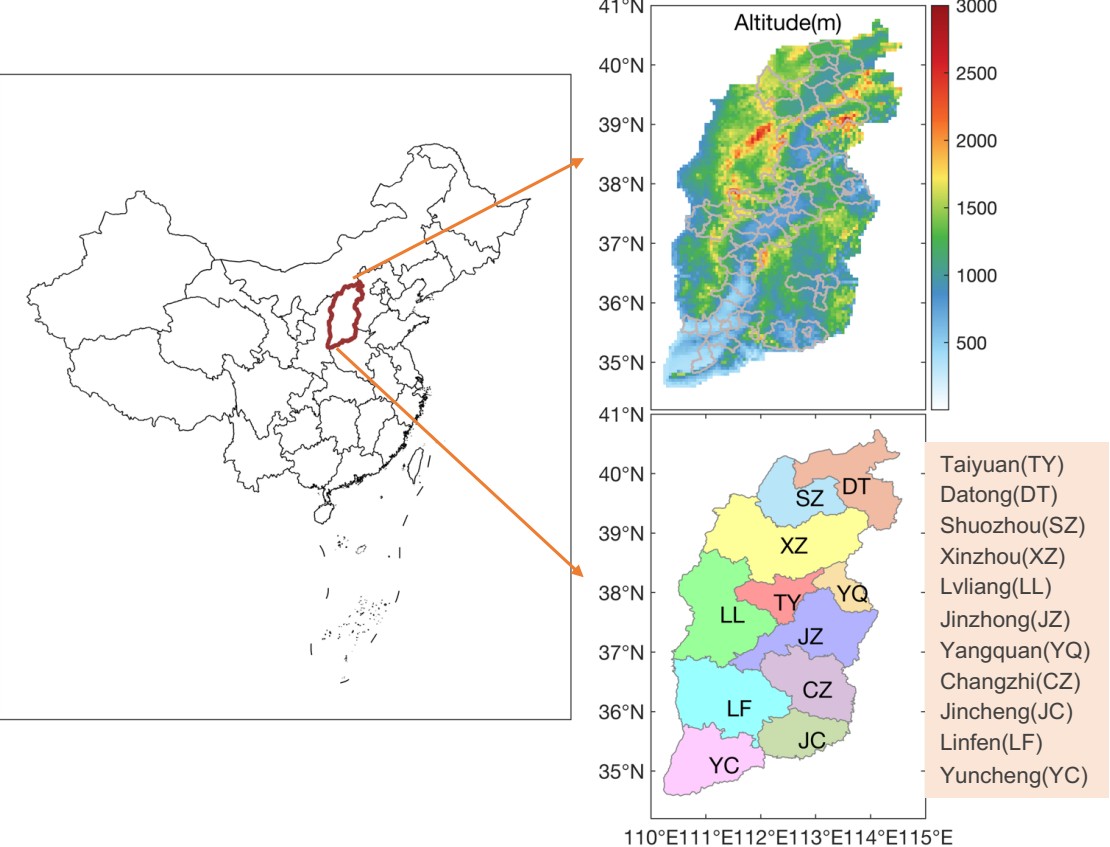

**Figure 1:** Location, topography and administrative division of Shanxi Province.

## 2 Materials and methods

### 2.1 Tropospheric vertical column measurements from TROPOMI

TROPOMI measures reflected solar radiation in the UV, visible, and Near IR bands following a sun-synchronous, low-earth orbit with an equator overpass time of approximately 13:30 LT, allowing daily-scale measurements across the globe (Veefkind et al., 2012; Goldberg et al., 2019; Tu et al., 2022a). Starting from August 2019, the spatial resolution of TROPOMI has been refined to 5.5 km×3.5 km (Lange et al., 2022). This study uses two distinct products measured by TROPOMI over different radiative bands, but at the same place and time: $NO_2$ and Ultraviolet Aerosol Index (UVAI).

Daily level-2 version 2.3.1 tropospheric $NO_2$ columns and version 2.2.0 UVAI over Shanxi Province has been introduced. All available days and swaths corresponding to the time period from January 2019 through December 2021 are analyzed (https://disc.gsfc.nasa.gov/datasets). Overlapping $NO_2$ and UVAI column pixels in each swath are resampled to a common latitude-longitude grid at 0.05 °×0.05 ° using weighted polygons (http://stcorp.github.io/harp/doc/html/index.html). Before use, it is required that all TROPOMI data is quality assured, specifically insisting that each pixel has a "qa_value" greater than 0.75, that the "cloud radiance fraction" is smaller than 0.5, and that scenes covered by snow/ice, errors and similar problematic retrievals are removed (Henk et al., 2021). Furthermore, an additional filter is applied to set all individual gird of $NO_2$ column which is less than $1.4×10^{15}$ molec cm$^{-2}$ to be NaN. This is done to avoid issues where the observed signal may be smaller than the uncertainty of the signal itself (J.H.G.M. Van Geffen, 2021; Qin et al., 2022). This combination of assumptions ensures that the data used should be of the highest possible precision based on the current available technology.

The average loadings, daily variation, and number of days without TROPOMI $NO_2$ columns from 2019 through 2021 are shown in Fig. 2. The number of invalid days varies pixel-by-pixel from 357 to 742 d (521 d on average), with higher altitude and mountainous areas tending to be more. The higher values on the maps are consistent with known urban and industrial regions, such as Taiyuan Basin, Xinding Basin, Linfen Basin, and Yangquan City. Areas with a high variation and a relatively low mean value are observed in regions where new economic development zones have been recently created or are in the process of being actively developed, including urban areas of Datong and Xinzhou. Areas with a relatively high variation and a high mean value are indicative of high urbanization and developed industrial areas, corresponding with the Taiyuan Basin, and southern Yangquan. Areas with a high average value and a low variation correspond with regions that have a fewer number of temporally consistent emission sources, as is observed in parts of the Linfen Basin, central Changzhi, Lvliang and Jincheng, and industrial parks in the Jiexiu district of Jinzhong.

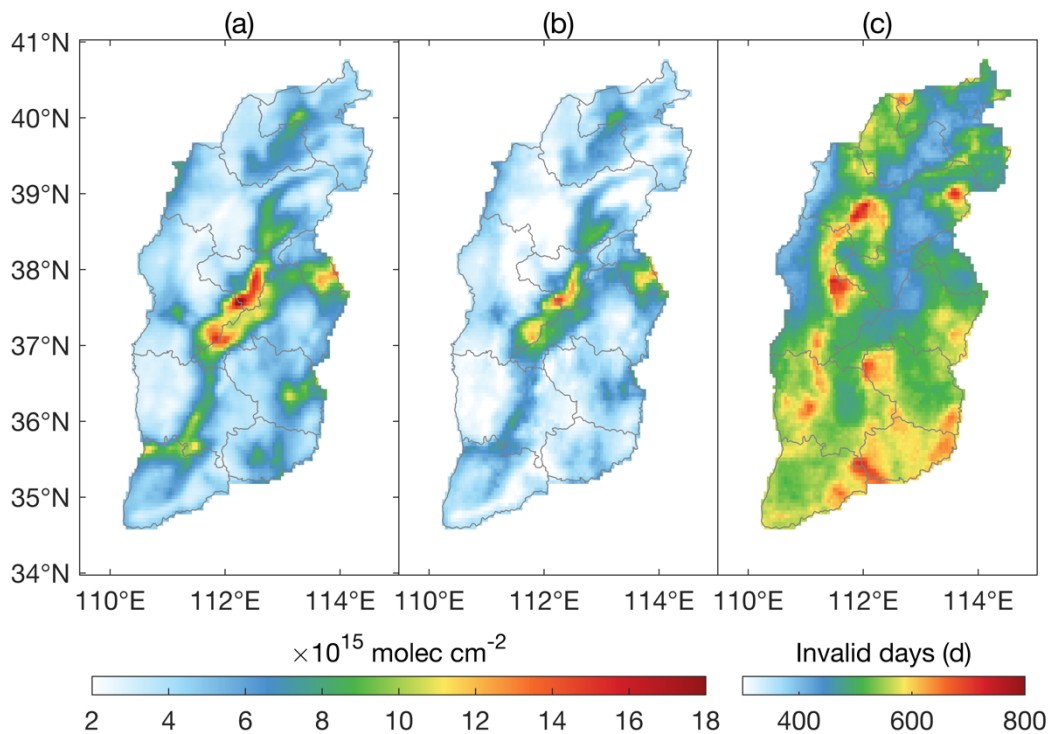

**Figure 2:** TROPOMI daily $NO_2$ column loadings from 2019 through 2021: (a) mean values (unit: molec $cm^{-2}$), (b) day by day standard deviation (unit: molec $cm^{-2}$), and (c) the number of invalid days (unit: d).

## 2.2 A prior emission inventories

### 2.2.1 CEMS

CEMS was introduced by the Ministry of Environmental Protection of China in 2007 to monitor and manage the emissions of certain (mainly high-emitting) plants (Schreifels et al., 2012; Karplus et al., 2018). CEMS makes actual stack flue gas measurements of the concentration of PM, $SO_2$ and $NO_x$ as emitted from power plants, iron and steel plants, aluminum smelters, coke plants, coal-fired boilers and others, all in real-time (Tang et al., 2020; Zhang and Schreifels, 2011). Statistics of the emissions sites monitored in Shanxi are given in Table 1. There are two different technologies for measuring $NO_x$ concentrations. One converts $NO_2$ to NO and measures the total NO concentration, the other measures $NO_2$ and NO separately. Both of the measured results have been converted to $NO_2$ mass concentration.

In this work, all available CEMS monitors of daily-scale emissions from 2019 to 2021 were obtained from the Department of Ecology and Environment of Shanxi Province (DEESP), with the government making great effort to regulate the CEMS network and to ensure the reliability of CEMS data (Tang et al., 2020). Preprocessing of these data include using google earth to correct the location of the factories, conducting quality control on measured concentrations according to the CEMS technical requirements, including eliminating negative values, outlier values, and null values. The overall percentage of abnormal values

is found to account for 0.63%, 1.18%, and 1.55% of the raw data respectively for 2019, 2020 and 2021. The formula used to calculate NO$_x$ emissions is given in Eq. (1):

$$E_d = \overline{C_h} \times \overline{Q_h} \times 24 \tag{1}$$

where $\overline{C_h}$ is the daily average of hourly NO$_x$ concentration, mg m$^{-3}$; $\overline{Q_h}$ is the daily average of hourly wet flue gas flow under actual working condition, m$^3$ h$^{-1}$, and 24 is used to convert units from hours to days. Following the "Specifications and test procedures for a continuous emission monitoring system for SO$_2$, NO$_x$, and particulate matter in flue gas emitted from stationary sources (HJ/T76-2017)", the uncertainty of NO$_x$ concentration ($C_h$) is less than or equal to 30% for the data used in this study. After quality control, the emission intensity on a grid-by-grid basis is found to be 0.64±0.08 µg m$^{-2}$ s$^{-1}$, 0.45±0.13 µg m$^{-2}$ s$^{-1}$, and 0.41±0.05 µg m$^{-2}$ s$^{-1}$ for 2019, 2020 and 2021, respectively. Probability density functions (PDFs) of daily emission intensity and a map of the multi-year mean are displayed in Fig. 3. The proportion of low values is the highest in one or two months of 2020, which are closely related to the epidemic, during which many key enterprises spontaneously shut down or otherwise limited production. However, on a year-by-year basis, the values in 2021 are lower than in 2020, reflecting the fact that the long-term efforts to reduce emissions may be even more important. Similarly, the highest emission intensity is observed in 2019.

**Table 1:** Summary statistics for plants included in CEMS.

| Year | Number of companies | Number of stacks monitored | Number of days without public data | Percentage of days without public data (%) |
|---|---|---|---|---|
| 2019 | 624 | 1607 | 102 | 27.9 |
| 2020 | 705 | 1819 | 24 | 6.6 |
| 2021 | 714 | 1836 | 0 | 0 |

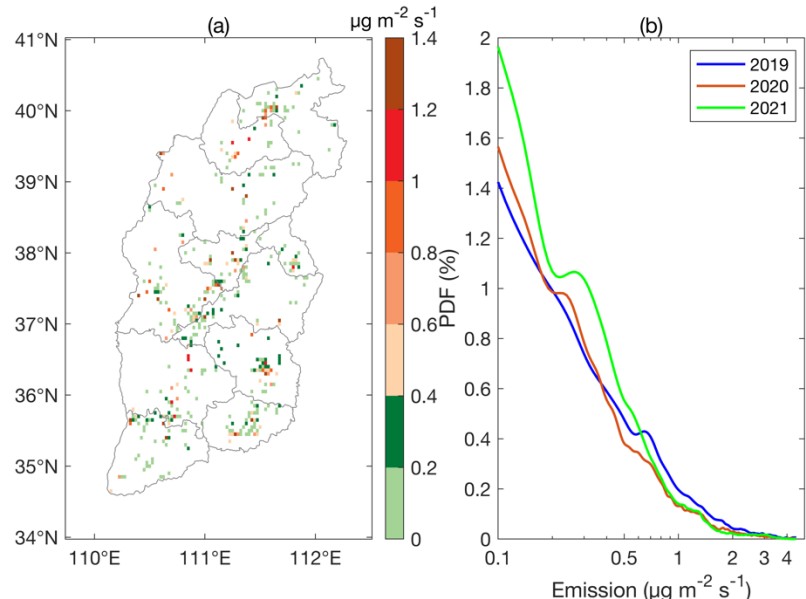

**Figure 3:** CEMS emissions intensity from January 2019 through December 2021 (unit: μg m$^{-2}$ s$^{-1}$): (a) 3-year average gridded NO$_x$ emissions, and (b) PDFs of day-by-day and grid-by-grid emissions over individual years (log-scale for x-axis).

### 2.2.2 MEIC

At the present time, some of the emission inventories most widely used by the community are MEIC (Zheng et al., 2018) and the Emission Database for Global Atmospheric Research (EDGAR) (Crippa et al., 2018). In this work, MEIC is selected since

it provides bottom-up emissions of anthropogenic air pollutants over mainland China, with a monthly time step and a 0.25°×0.25° spatial resolution. NO$_x$ emissions are provided from January 2019 to December 2020 over five sectors: agriculture, industry, power, residential and transportation (Zheng et al., 2021). To match with the higher resolution of TROPOMI grids, all of the MEIC data in this work is mapped uniformly to a 0.05 °×0.05 ° grid, with each TROPOMI sized grid assigned the same flux as the underlying MEIC grid. The average emission values and PDFs of the grid-by-grid data over Shanxi from

2019 January to 2020 December are given in Fig. 4a and Fig. 4b, demonstrating little change in emissions between the two years except for at very low values.

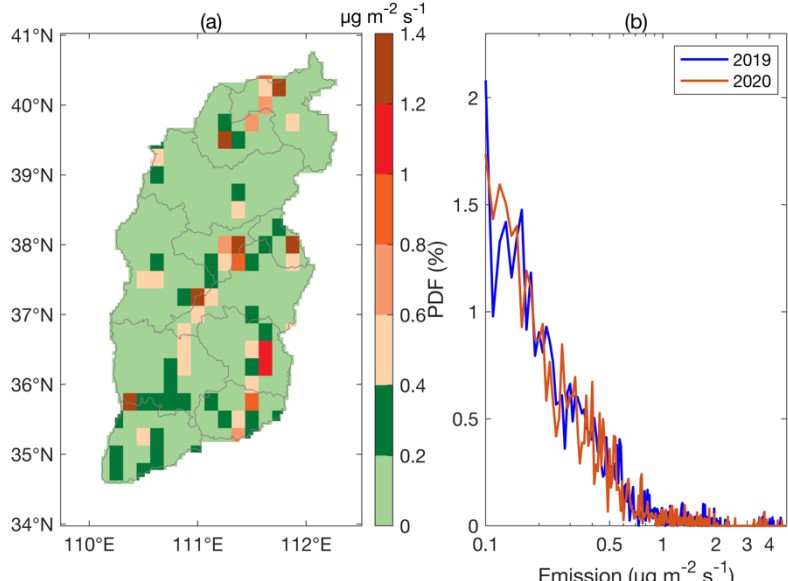

**Figure 4:** MEIC emissions intensity from January 2019 through December 2020 (unit: μg m$^{-2}$ s$^{-1}$): (a) 2-year average monthly MEIC emissions, and (b) PDFs of month-by-month and the grid-by-grid emission over individual years (log-scale for x-axis).

**2.3 Wind**

Wind speed and direction are from the European Centre for Medium-Range Weather Forecasts, ERA-5 reanalysis products. This work uses the hourly 6:00 UTC u and v wind products (closest in terms of time to the TROPOMI overpass) at 850 hPa and 0.25° × 0.25° resolution, available at https://www.ecmwf.int/en/forecasts/dataset/ecmwf-reanalysis-v5. The wind was linearly interpolated to follow the TROPOMI 0.05° × 0.05° grid data in space. The reason for choosing the 850 hPa (which is

200 approximately 1500 m) level is two-fold. First, Shanxi has complex topography, with less than 16% of the total area of the

province under 800 m in height, 68% of the surface over 1000 m, and 17% of area over 1500 m, see Fig. 1, leading to a significant amount of pollutant transport from near the ground to the lower free troposphere. Second, due to the relatively dry conditions, vertical plume-based rise is thought to not be insignificant (Wang et al., 2020). Overall, we aim to use wind speed and direction that correspond to a reasonable approximation of the median of the $NO_x$ emissions vertical profile.

## 2.4 Variance maximization

To extract the spatial and temporal features of the extremes of the remotely sensed $NO_2$ fields in an unbiased manner, the Empirical Orthogonal Functions Principal Components Analysis (EOF) is applied. This technique decomposes the data into a set of orthogonal standing signals in space [EOF] and in time [PC], with those signals contributing the most to the overall variance of the underlying dataset being selected, representing unique phenomenon that control the overall characteristics of the $NO_2$ columns (Zhou et al., 2016; Lin et al., 2020). Further details including mathematical derivations are given in Björnsson and Venegas (1997) and Cohen (2014). This work retains the first three EOFs, which are found to contribute to 43.3%, 6.4%, and 3.9% of the total variation, with subsequent EOFs each contributing an insignificant amount (less than 3.9%) and therefore no longer considered in this work.

## 2.5 Model Free Inversion Estimation Framework (MFIEF)

The MFIEF is based on a mass balance assumption Eq. (2), and the detail is shown in Fig. 5. In the case where there is an observed change in the stock of $NO_x$ in the atmosphere, herein represented as $C$, in a Lagrangian sense, there must be either a source or sink (Harte, 1988; Seinfeld and Pandis, 1997). When dealing with a fixed spatial grid, such as in this work, there is also a contribution of transport into or from the Lagrangian parcel. The first of these changes in the stock is due to the amount of $NO_x$ emitted, herein represented as $E$, which will always increase the existing stock. The second of these is the chemical loss of $NO_x$, which will always lead to a decrease in the stock. The chemical sink of $NO_x$ is dominated by the reaction between $NO_2$ and OH, via reactions with products formed from the actinic flux (i.e., chemistry such as $RO_2$), and on aerosol surfaces via heterogeneous reactions (Valin et al., 2013; Kenagy et al., 2018; Romer Present et al., 2020), which herein is described as S. The third change in the stock is the sum of pressure induced and advective transport, which may either increase or decrease the stock. The transport is herein is described as $D$, and is calculated by the gradient of the product of the wind vector and the $NO_x$ column loadings, which consists of an advective portion (Wang et al., 2014) and a pressure-based portion (Mahowald et al., 2005). The mass conservation equation for $NO_x$ is calculated as

$$dC = E - S + D \tag{2}$$

Solving Eq. (2) for emissions on a grid-by-grid basis requires knowledge of the mass change of the loading in time and space, and detailed consideration of chemical loss and transformation, and transport. An explicit formulation of these processes into a readily solvable mass balance method is derived as Eq. (3):

$$E_{NOx} = \alpha_1 \cdot \frac{dV_{NO_2}}{dt} + 24 \cdot \frac{\alpha_1}{\alpha_2} \cdot V_{NO_2} + 0.001 \cdot \frac{\alpha_1}{\alpha_3} \cdot (\nabla(\boldsymbol{u} \cdot V_{NO_2}) + \nabla(\boldsymbol{v} \cdot V_{NO_2})) \qquad (3)$$

Where $E_{NOx}$ represents the total atmospheric column emissions of $NO_x$ within the troposphere on a grid-by-grid and day-by-day basis, with a unit of $\mu g\ m^{-2}\ d^{-1}$. This is the total emission into each column accounting for all sources including anthropogenic sources (industry, vehicle, and residential), biomass burning, and others. $V_{NO2}$ represents the tropospheric $NO_2$ column concentrations after it has been converted into the unit of $\mu g\ m^{-2}$. Due to the fact that TROPOMI only measures $NO_2$ rather than $NO_x$, a transformation is required to transform $NO_2$ columns into $NO_x$, hereafter given as $NO_x = \alpha_1 \cdot NO_2$. $\alpha_1 \cdot \frac{dV_{NO_2}}{dt}$ computes the $NO_x$ concentration change rate with a unit of $\mu g\ m^{-2}\ d^{-1}$, assuming the day-to-day temporal derivative of $NO_2$ exists in the TROPOMI data on the respective days. $24 \cdot \frac{\alpha_1}{\alpha_2} \cdot V_{NO_2}$ represents as the sink of $NO_x$ concentration, where $\alpha_2$ is related to the $NO_x$ lifetime (unit h), and 24 is the unit change factor. $\nabla(\boldsymbol{u} \cdot V_{NO_2}) + \nabla(\boldsymbol{v} \cdot V_{NO_2})$ represents the gradients of daily zonal fluxes, meridional fluxes, and changes in the air column mass and density. These were derived by multiplying the gridded $V_{NO2}$ with the center point wind vector on a grid-by-grid basis, with unit of $\mu g\ m^{-2}\ d^{-1}$. This computation assumes that the spatial gradient of TROPOMI $NO_2$ and reanalysis wind both exist on the respective grids (Cohen and Prinn, 2011). In this case, $\alpha_3$ is the parameter representing the transport distance (km) per day, where 0.001 is used to convert m to km. $V_{NO2}$ is observed as either one or two overlapping snapshots of total column information occurring at 13:30 LT (and under some conditions also 101.5 minutes either earlier or later (Tonion and Pirotti, 2022)). In all cases, the meteorological values and CEMS values are representative of 24-hour total and/or daily average conditions respectively. Therefore, the fitted values of $\alpha_1$, $\alpha_2$, and $\alpha_3$, as presented are representative of 24-hour average or 24-hour net effect respectively, acting on the entire column of $NO_x$. In all cases, the values of $\alpha_1$, $\alpha_2$, and $\alpha_3$ are computed month-to-month over all grids in which data is available, and by bootstrap at other grids and times.

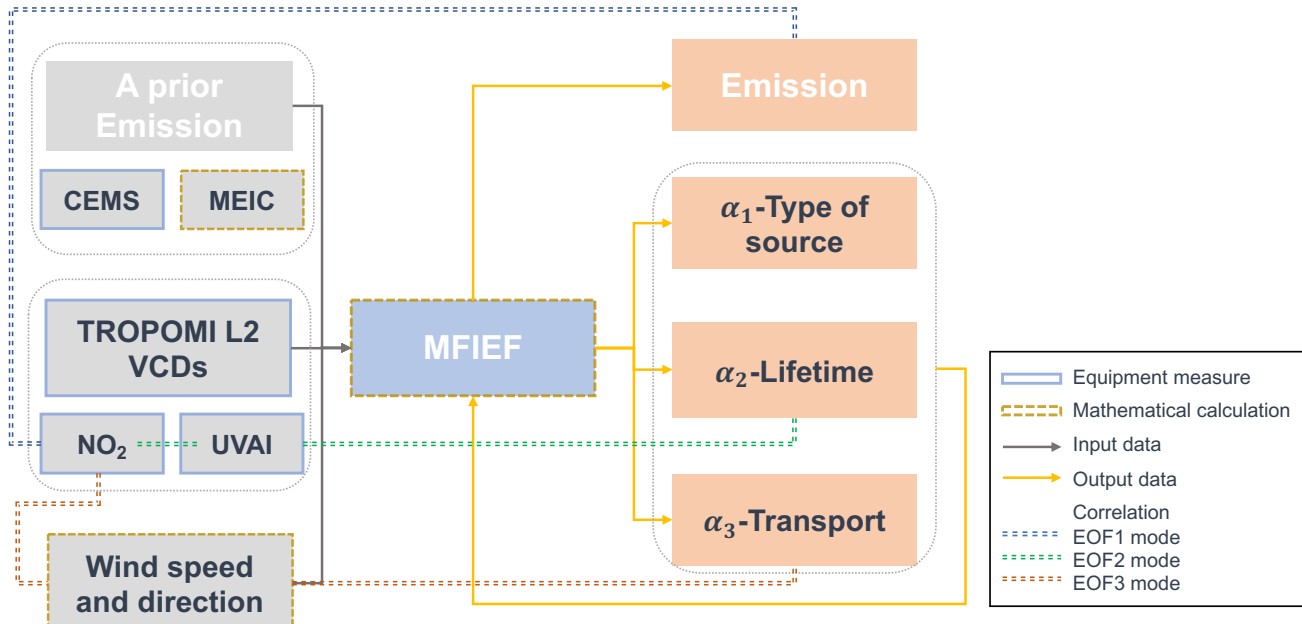

**Figure 5:** The framework of MFIEF methodology.

**2.6 Additional analytical methods**

This work employs multiple linear regression to fit the values of $\alpha_1$, $\alpha_2$, and $\alpha_3$ on a month-by-month, grid-by-grid basis using all available daily measurements and Eq. (3). These values are further filtered based on both statistics (p<0.15, and removal of
255 outliers defined as elements more than three scaled MAD from the median, calculated by Eq. (4)) and being physically realistic ($|\alpha_1|>1$, $\alpha_2<0$).

$$MAD = c \times median(abs(\ \alpha - median(\alpha))) \tag{4}$$

$$c = \frac{-1}{\sqrt{2} * erfcinv(\frac{3}{2})} \approx 1.483$$

$\alpha_1$, $\alpha_2$, and $\alpha_3$ are brought into $\alpha$ in Eq. (4) respectively for outlier rejection. Bootstrapping is used as a means to create a new
sample to represent the parent sample distribution through multiple repetitions of sampling (Liu and Cohen, 2022). Specifically, the distributions of $\alpha_1$, $\alpha_2$, and $\alpha_3$ are sampled across the central 80% of their probability distributions, which are then used to generate a set of pseudo $\alpha_1$, $\alpha_2$, and $\alpha_3$ on a grid-by-grid basis where there is no existing a priori and therefore no actual solution of these variables. These bootstrapped pseudo $\alpha_1$, $\alpha_2$, and $\alpha_3$ are then used on these specific grids to approximate the emissions of $NO_x$ using Eq. (3) on a daily basis where TROPOMI $NO_2$ column data and wind data is available. The mean
data has been taken out as the daily emission, while the standard deviation was calculated as the uncertainty of the daily emission.

# 3 Results and discussion

## 3.1 Computed emissions using CEMS and MEIC

Fig. 6a and Fig. 6b show the spatial distribution of daily average and variation of $NO_x$ emissions based on CEMS ($EI_{CEMS}$) from January 2019 to December 2021 over Shanxi at 0.05° × 0.05°. For all subsequent emissions values displayed, the numbers correspond to the sum over the day-to-day mean ± uncertainty. It is observed that the grids with the highest $NO_x$ emission in Shanxi are mainly concentrated in the lower Fen River valley, which happens to also be located at the lowest altitude areas in the province as shown in Fig. 1 pink line area containing Taiyuan Basin, Xinding Basin, and Linfen Basin, which also corresponds to the area containing the highest population density. This area in total accounts for around 25% of the total area of the province and contributes 53% (0.98±0.55 Tg per year) of the total $NO_x$ emission (1.86±1.05 Tg per year) in Shanxi. It is of significance to note the regions with a moderate average density of emissions, ranging from 0.3 to 0.7 µg m$^{-2}$ s$^{-1}$, contribute 62% (1.15±0.64 Tg per year) of the total emissions. The use of MFIEF can effectively optimize the distribution of the inventory and perform inventory correction based on satellite data, while complementing many areas where there is no existing emissions data, incomplete data, mis-characterized data, or data which may be reasonable on average but not account for daily-scale variability.

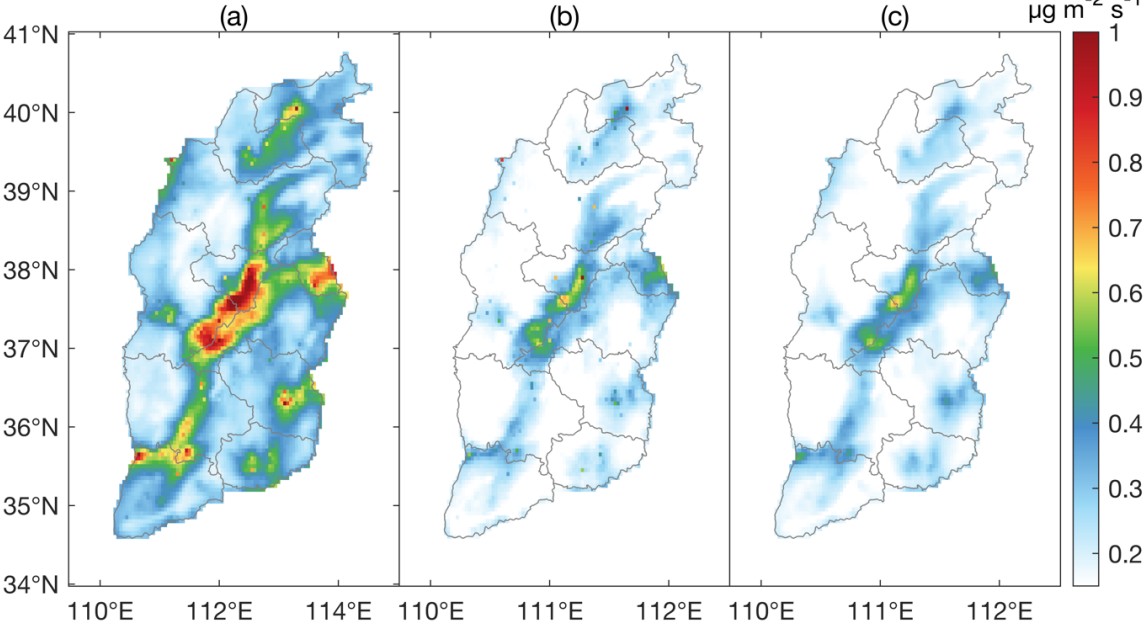

**Figure 6:** Daily average emissions based on CEMS from January 2019 to December 2021 over Shanxi at 0.05° × 0.05° (unit: µg m$^{-2}$ s$^{-1}$): (a) daily average $NO_x$ emissions. (b) day-to-day variability of $NO_x$ emissions. (c) bootstrapping uncertainty range (10% to 90% of distribution).

MEIC was also separately used as the a priori with MFIEF to produce an additional emissions inventory from January 2019 to December 2020, herein called $EI_{MEIC}$. The total $EI_{MEIC}$ over the province is 1.34±0.60 Tg per year. The multi-annual mean value of $EI_{CEMS}$ minus $EI_{MEIC}$ and histogram of the day-to-day and grid-to-grid emissions of both $EI_{CEMS}$ and $EI_{MEIC}$ from 2019

to 2020 are displayed in Fig. 7. On a multi-annual average basis, EI$_{CEMS}$ is larger than EI$_{MEIC}$ at 98% of grids, while on a day-to-day basis it is larger at 90% of grids. Some higher value appeared in EI$_{MEIC}$ calculated by high MEIC values in some places of Shuozhou, Yangquan, Lvliang, Changzhi, and Jincheng may be due to mis-positioned hotspots in the existing inventories, in terms of both space and time. But on the opposite, EI$_{CEMS}$ better captured in residential and small industrial sources than EI$_{MEIC}$. This is possibly due to the fact that the a priori MEIC distribution has many low values which actually fall within the uncertainty of the bottom-up emissions processes (Cohen and Wang, 2014; Bond, 2004; Crippa et al., 2018). The low a priori emissions on a grid-by-grid basis, which in turn shift the physically filtered values of $\alpha_1$ towards lower values. At the same time the values of $\alpha_2$ were shifted to higher values. In tandem with these, the transport term $\alpha_3$ was shifted to include larger absolute values of distance. This net combination caused the final calculation to be biased smaller overall.

Fig. 8 shows the differences between CEMS, EI$_{CEMS}$, EI$_{MEIC}$ and MEIC, and the 30% error range for CEMS and the computed 10[th] to 90[th] percentile error ranges for EI$_{CEMS}$ and EI$_{MEIC}$ as computed from January 2019 to December 2020 in all 11 cities in Shanxi. It shows that while generally CEMS is larger than EI$_{CEMS}$, that they are always found to be within the error ranges of each other, while EI$_{MEIC}$ only overlaps with CEMS in 6 cities. Similarly, MEIC is always found to be the lowest, while EI$_{MEIC}$ is found to be larger than MEIC and smaller than EI$_{CEMS}$. In specific EI$_{CEMS}$ and EI$_{MEIC}$ have error ranges which overlap in all cities, while MEIC overlaps with the EI$_{MEIC}$ range in 8 cities.

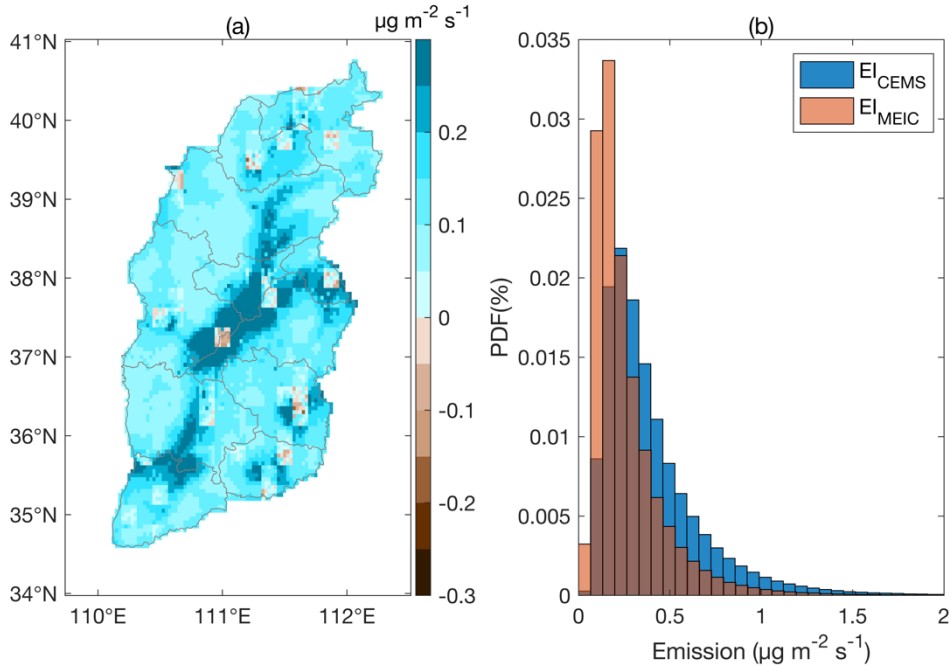

**Figure 7:** Differences between EI$_{CEMS}$ and EI$_{MEIC}$ through 2019 to 2020 (unit: $\mu g\ m^{-2}\ s^{-1}$): (a) average values in space of EI$_{CEMS}$ minus EI$_{MEIC}$. (b) Histogram of EI$_{CEMS}$ and EI$_{MEIC}$.

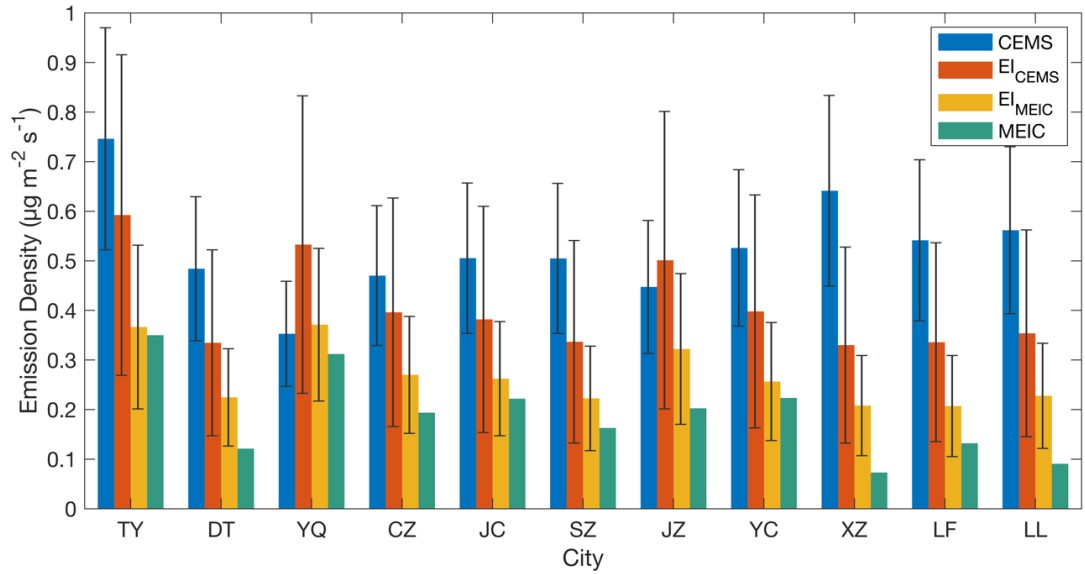

**Figure 8:** Differences between CEMS, $EI_{CEMS}$, $EI_{MEIC}$ and MEIC with their error ranges through 2019 to 2020 in all 11 cities in Shanxi (unit: $\mu g\ m^{-2}\ s^{-1}$).

### 3.2 Underlying factors contributing to variance maximized TROPOMI NO₂ columns

A deeper analysis of the factors contributing to the variance in the TROPOMI NO₂ column measurements is essential to
310 determine if the computed emissions and underlying factors are consistent with the remotely sensed fields both in terms of mean value and temporal variability. Recent practice has devised a way to ensure this consistency through the use of an EOF Analysis (Cohen et al., 2017; Cohen, 2014; Lin et al., 2020), which is applied to the daily TROPOMI NO₂ columns. The three spatial modes contributing the most variation to the observed daily TROPOMI NO₂ fields [EOF1, EOF2, and EOF3] contribute 43.3%, 6.4%, and 3.9% respectively, as shown in Fig. 9.

It is asserted that EOF1 is directly driven by $EI_{CEMS}$. The comparison of EOF1 and the emissions is shown in Fig. 10 in terms of both spatial and temporal scales. By applying four different progressively increasing cutoffs to the domain of EOF1, it is observed that as the EOF1 domain increases in magnitude, that the 3-year mean $EI_{CEMS}$ computed over the same domains also increase in magnitude. Therefore, the more extreme the EOF1 value, the higher the emissions, demonstrating that the emissions are responsible for the first mode of the maximized variance. $EI_{CEMS}$ is also found to statistically correlate with PC1 ($r$=0.60,
p<0.01), indicating that the first mode of variation is strongly connected with the changes in $EI_{CEMS}$ in terms of both spatial and temporal dimensions.

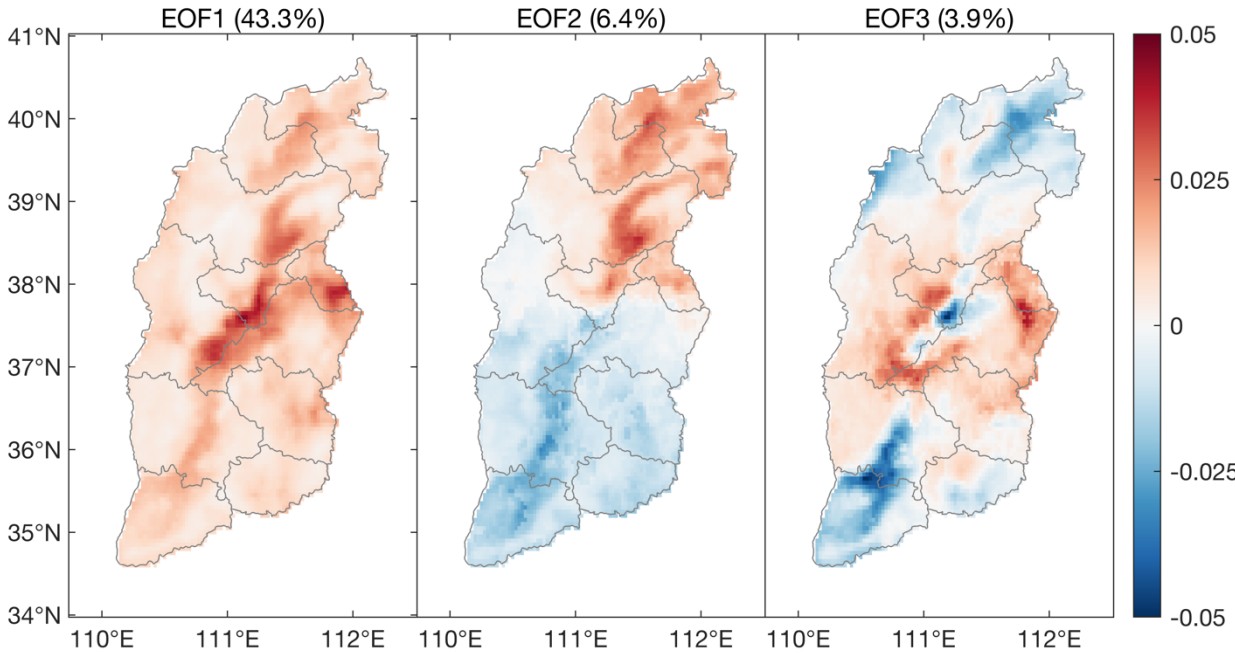

**Figure 9:** Spatial distribution map of first three modes (a) EOF1, (b) EOF2, and (c) EOF3.

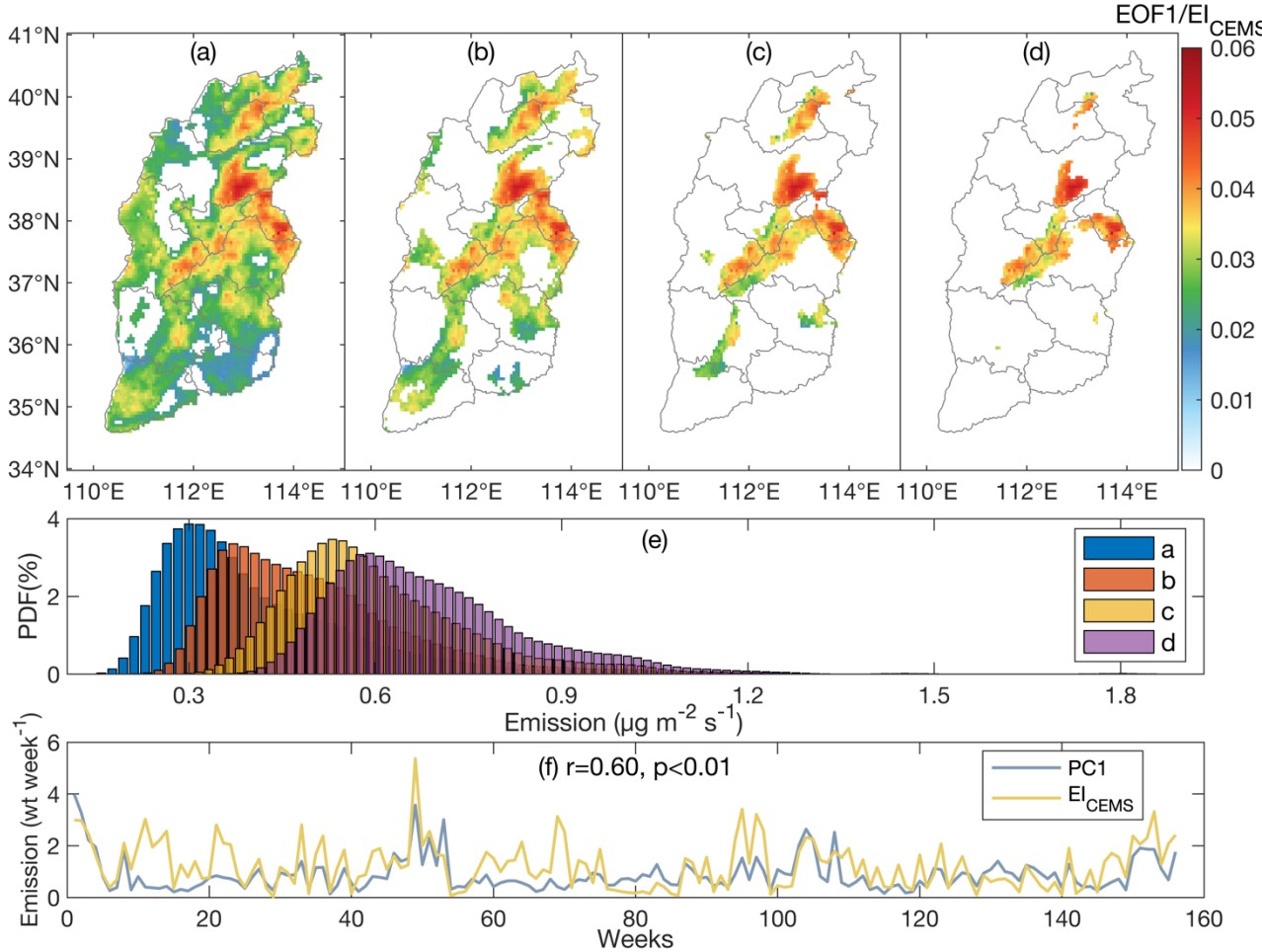

**Figure 10:** Four different cutoffs of EOF1 are used to set the different spatial domains. The maps in (a-d) are plots of EOF1/EI$_{CEMS}$ where the cutoffs are given as (a) EOF1 >0.005, (b) EOF1 >0.01, (c) EOF1 >0.015, (d) EOF1 >0.02. (e) Histograms of the EI$_{CEMS}$ over the domains given respectively in a-d. (f) Time series of weekly PC1 compared with EI$_{CEMS}$ over the whole domain.

Second, it is asserted that EOF2 is related to TROPOMI measured UVAI, which physically makes sense, since satellite observations of UVAI are sensitive to aerosol extinction in the UV, with very large values indicating large amounts of highly absorbing aerosol (SSA less than 0.9) and very large negative values indicating large amounts of partially absorbing aerosol (SSA between 0.92 and 0.98) (Penning et al., 2009; Torres et al, 2020). There have been numerous studies reporting that absorbing aerosols affect the downwelling surface radiative forcing in the visible (and therefore the actinic flux) (Léon, 2002) as well as OH concentrations (Hammer et al., 2016). Therefore, UVAI is an indirect proxy of an aspect contributing to the chemical decay capacity of NO$_x$ in-situ. Applying four different cutoffs to EOF2, it is observed that as the EOF2 domain increases in magnitude, that the 3-year mean measured absolute value of TROPOMI UVAI becomes smaller in magnitude, as demonstrated in Fig. 11. Since values of UVAI closer to zero imply less atmospheric extinction (absorption for positive values and a mixture of scattering and absorption for negative values), it therefore also scales inversely with surface UV radiation,

implying that when the UVAI is lower, that there is more available UV radiation, and hence implicitly faster chemical decay of NO$_x$. This is consistent with the UV radiation being responsible for the second mode of the maximized variance. The negative correlation observed between absolute values of UVAI weighted by high absolute values of EOF2 (EOF2>0.02) grid-by-grid in the same EOF2 region is anticorrelated with PC2 ($r$=-0.33, p<0.01). While $r$ in this case is a lot smaller, it is also consistent with theory, since in order to significantly affect the OH levels, the changes in UV radiation and hence UVAI must be very large, which is found to not occur frequently in time, but when it does occur, it makes a significant impact.

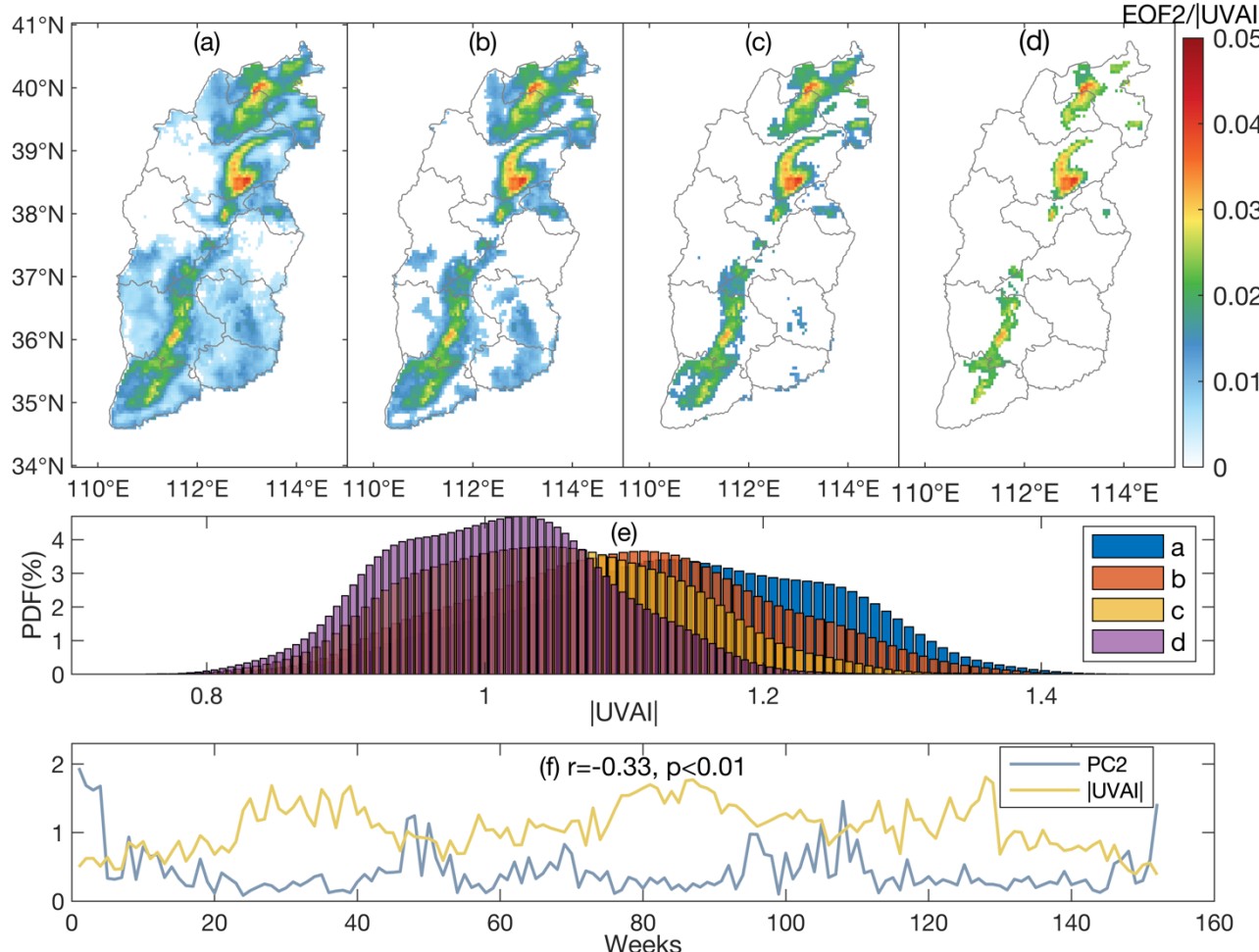

**Figure 11:** Four different cutoffs of EOF2 are used to set the different spatial domains. The maps in (a-d) are plots of EOF2/|UVAI| where the cutoffs are given as (a) EOF2 >0.005, (b) EOF2 >0.01, (c) EOF2 >0.015, (d) EOF2 >0.02. (e) Histograms of |UVAI| over the domains given respectively in a-d. (f) Time series of weekly PC2 and |UVAI| in the (d) domain.

Finally, it is asserted that EOF3 is related to the transport of NO$_2$. This term has been specifically computed by taking the variance of the multiple of wind and TROPOMI NO$_2$ column loadings, specifically $\nabla(\boldsymbol{u} \cdot V_{NO2})$. Similarly, to the above cases, it is demonstrated that as four different cutoffs are applied to EOF3, it is observed that as the EOF3 domain increase in

magnitude, so does the measured transport based on TROPOMI NO$_2$ also increase, as observed in Fig. 12. $\nabla(\boldsymbol{u} \cdot V_{NO2})$ weighted by EOF3 grid-by-grid positively correlates with PC3 ($r$=0.74, p<0.01), indicating that the third mode of variability, PC3 is strongly consistent with $\nabla(\boldsymbol{u} \cdot V_{NO2})$ in the temporal dimensions. Therefore, transport is responsible for the third mode of the maximized variance.

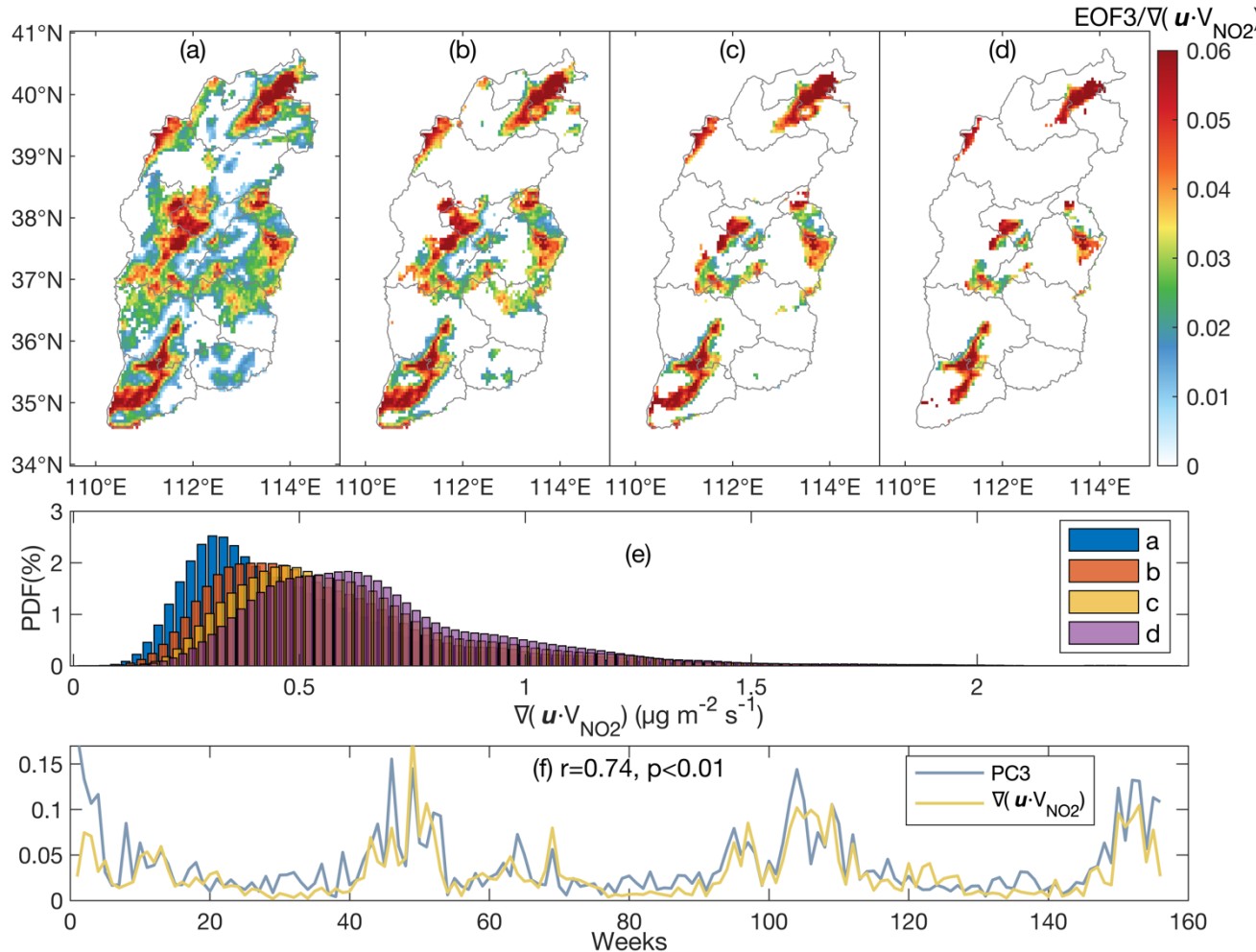

**Figure 12:** Four different cutoffs of EOF3 are used to set the different spatial domains. The maps in (a-d) are plots of EOF3/NO$_2$-transport where the cutoffs are given as (a) EOF3 >0.005, (b) EOF3 >0.01, (c) EOF3 >0.015, (d) EOF3 >0.02. (e) Histograms of the NO$_2$-transport over the domains given respectively in a-d. (f) Time series of weekly PC3 compared against $\nabla(\boldsymbol{u} \cdot V_{NO2})$ over the whole domain.

### 3.3 Uncertainty in emission and the parameters

The uncertainty of the computed emissions on a grid-by-grid and day-by-day basis is due to a combination of uncertainties in the satellite data, CEMS or MEIC as a priori emissions, and the calculations involved with computing the various best-fit parameters. NO$_x$ column concentrations and CEMS both have uncertainties about 30%. Another uncertainty comes from the parameters $\alpha_1$, $\alpha_2$, and $\alpha_3$ generated during the regression of Eq. (3) as given in section 2.5. The fitted coefficients are computed

month-by-month over the three years from January 2019 through December of 2021. Their absolute value overall mean, 10th

percentile, and 90th percentile are found to be $\alpha_1$=[4.0,1.3,8.2], $\alpha_2$=[12.3,7.1,18.1] h, and $\alpha_3$=[239,63,508] km. However, it is observed in the fits that some amount of the variability is not uniform in space and time, with the month-by-month values and standard deviations given in Fig. 13. In general, $\alpha_1$ tends to be slightly higher during one or a combination of both hotter months and times when the UVAI is low (hence the UV values are high). $\alpha_2$ shown the lifetime reflect all hours during the day. In general, it tends to be variable, both inter-annual and intra-annual variations seem to drive most of the change. Given

that this is related to both the column average temperature and UV availability, the largest values are found in June or July and the lowest values are found in September. Furthermore, there are other complex forcing factors including the height of the aerosol layer, the total aerosol loadings, cloudiness, and other factors. The absolute magnitudes of $\alpha_2$ and their uncertainty range are reasonable when compared with vertically integrated and 24-hour integrated chemical transport model values. In general, $\alpha_3$ also seems to not have any significant seasonal or monthly pattern, with inter-annual and intra-annual terms seeming

to dominate. The values tend to be slightly larger than chemical transport models account for, but are reasonable when compared with the ultra-long-range transport simulated for plumes which break the boundary layer. This range, combined with the wide basins of 200 km to 400 km in length, seem to provide a reasonable bound on the output results. One possible reason for this is that the atmospheric wind patterns were slightly different in 2019 due to the El Niño pattern (Hu et al., 2020). The result of overall uncertainty of the emissions as a result of the overall bootstrapped fitting ranges from 32% to 70% on a grid-

by-grid basis, as shown in Fig. 6c. In general, the larger relative uncertainty values are observed over mountain regions. A sensitivity test has been performed to test the robustness of the fits assuming that the TROPOMI $NO_2$ column values are varied near the extreme top and bottom of their 30% uncertainty range. It is shown that the new $NO_x$ emissions in the 70% case are always larger than 0.7 times the original emissions case (spatially annually averaged and grid-by-grid this varies from 0.89 to 1.00, while temporally domain averaged this varies from 0.77 to 1.04). Similarly, the new $NO_x$ emissions in the 130% case

over the median 90% of data is smaller than 1.3 times the original emissions case (spatially annually averaged and grid-by-grid this varies from 1.08 to 1.12 while temporally domain averaged this varies from 1.01 to 1.30).

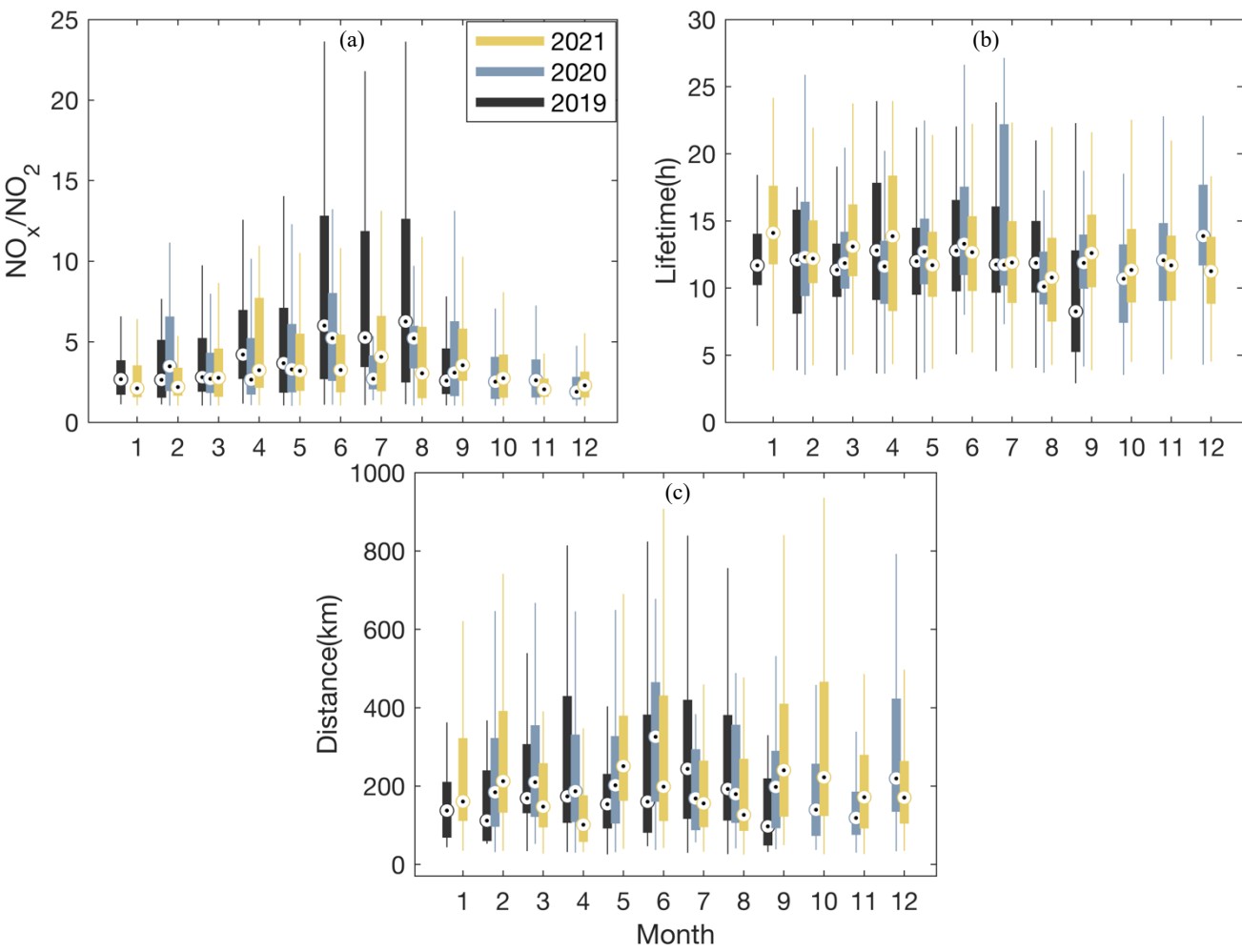

**Figure 13:** Distribution of monthly $\alpha_1$ (a), $\alpha_2$ (b) and $\alpha_3$ (c) calculated based on CEMS.

### 3.4 Application of $\alpha_1$ to analyze different combustion technologies

A significant finding is observed when the value of $\alpha_1$ is analyzed more closely on a pixel-by-pixel level and compared with underlying CEMS combustion source types. This analysis is motivated by the fact that $NO_x$ is produced during high temperature combustion of air, with three different major parts contributing to the overall amount of $NO_x$ produced: thermal $NO_x$ formation, fuel $NO_x$ formation and chemical $NO_x$ formation (Schwerdt, 2006; Le Bris et al., 2007). Thermal $NO_x$ formation describes the process when $N_2$ reacts with $O_2$ in the air at high temperatures (Le Bris et al., 2007), with $NO_2$ forming

preferentially at temperatures between 800 ℃ and 1200 ℃ and NO forming preferentially at temperatures above 1200 ℃. Thermal $NO_x$ usually dominates the overall $NO_x$ production when the temperature is over 1100 ℃, and reaches a maximum contribution when the temperature is over 1600 ℃. There is additional $NO_x$ produced due to free nitrogen in the fuel itself. Finally, chemical decay may occur when there are mixed organo-nitrides, resulting in the prompt $NO_x$ formation. Therefore,

a deeper understanding of the overall and oxygen partial pressures and temperature in the combustion chamber are all important for $NO_x$ formation. In summary, as the temperature of combustion increases, both the amount of NO and $NO_2$ will increase. When the temperature exceeds 1200 °C, NO will continue to increase while $NO_2$ will decrease. During any time when the pressure increases, the yield of $NO_2$ will also decrease and NO will increase (Aho et al., 1995; Turns, 1995).

A deeper look at the various different CEMS sources (cement, power, iron and steel, coke, boilers, and aluminium oxide) reveals that the internal combustion processes are extremely important in terms of the overall value of $\alpha_1$. The statistical distribution of $\alpha_1$ and the calculated values of $\alpha_1$ at the $10^{th}$, $25^{th}$, $50^{th}$, $75^{th}$, and $90^{th}$ percentile values (given in the Table 2) clearly demonstrated that cement factories have the highest value, power and iron and steel have the next highest values, coke and aluminum oxide are significant lower still, and boilers have the lowest values. Further details are also clearly observed, for example, power and iron/steel have significant differences across different parts of the distribution with power having larger values from the $25^{th}$ through $75^{th}$ percentile range and iron and steel having a higher value at the $90^{th}$ percentile level. This is consistent with the fact that iron and steel use several different processes, one of which contains very high temperature combustion (blast furnace based) and the other which contains a lower temperature process (sinter bed based). Other differences are explained more in depth herein.

Production of cement is a major source of $NO_x$ in Shanxi, with the major technology being dry process rotary kiln technology. Given that the temperature of the main burner of cement rotary kilns are higher than 1400 °C, with some peaking as high as 1800 °C to 2000 °C (Wu et al., 2020; Akgun, 2003), it is expected that there will be a large amount of thermodynamic $NO_x$ generation. As observed at the cement CEMS sites, the computed $\alpha_1$ has a value always within or above the values computed at power plants, including both in terms of the mode, as well as at the highest emitting individual values, as show in Table 2 and Fig. 14. Iron and steel are produced through a set of different processes, involving combustion at a range of different temperatures. The steps involved in the blast furnaces as well as some other processes, require a high flame temperature, in the range from 1350 °C to 2000 °C. There are further processes occurring that require a relatively lower temperature, such as in the sinter bed stage, where the highest temperature is only about 1300 °C (Zhou et al., 2018). Therefore, while in general the values are relatively high compared to the other source types below, and are generally found within the range of values at power plants, there are some individual differences as well, including a larger fraction of large values (9-10) and a smaller fraction at very large values (15-17), as observed in Fig. 14. The maximum temperature of the combustion chamber of thermal power plants can reach 2000 °C. In fact, such plants are constantly finding ways to increase the combustion efficiency, so that they can be more energy efficient and produce as much energy per ton of $CO_2$ emitted. $\alpha_1$ is relatively high at these sites, consistent with thermal production.

Boilers use a similar technology as power plants, but tend to be smaller and run at a lower temperature range and efficiency. This is because their use is to produce hot water and steam for direct residential and industrial use, not high-pressure steam to run turbines. In general, these boilers have a much smaller overall capacity and therefore without access to CEMS, may not be otherwise be detectable. For these reasons, it is logical that there is a greater amount of $NO_2$ produced than the above cases, and subsequently the value of $\alpha_1$ is much lower, in terms of both the mode, as well as all moderate values (5 and larger) as

shown in Table 2 and Fig. 14. Coke and aluminum oxide are both produced using a different technique from the other combustion sources, specifically focusing on creating high temperature, oven-like conditions to bake/roast their products. The average temperature of the coke oven charring chamber and aluminum oxide roasting furnace are around 1000 °C (Abyzov, 2019; Neto et al., 2021), with the material temperature continuously held in that temperature for a long period of time, e.g., one day. At the same time, the oxygen content is low. In net, there is far less thermal NO and more thermal $NO_2$. Aluminium is also smelted in an oven-like condition. Correspondingly, while the mode of $\alpha_1$ for aluminium oxide is more similar to iron and steel, and the mode of $\alpha_1$ for coke is slightly lower, their distributions over the range are quite different from each other. Iron and steel is consistently higher than the other two from the 25[th] percentile and upwards, aluminium is higher than coke from the 50[th] percentile and lower, coke is higher than aluminium from the 75[th] percentile and higher. All of the species have distributions in which the bulk of their distribution are skewed lower, while simultaneously exhibiting a small but not insignificant tail, as detailed in Fig. 14.

In addition, in-situ processes also impact the value of $\alpha_1$ since there is a rapid adjustment after emitted from a combustion stack into the atmosphere, before the parcel comes to thermodynamic equilibrium (Cohen et al., 2018; Wang et al., 2020). From Fig. 13a it can be seen while there is a pattern where the values are higher in certain months than others, that there is also a clear difference between the different source types. This demonstrates clearly that the atmospheric processing modulates the process to some extent, but does not dominate the signal on average. This is especially important in the case of hotter power sources, since they will contain more buoyancy, and rise to a higher height, making them more likely to be in contact with air which is more exposed to UV and also generally colder than the surface. Overall, the value of $\alpha_1$ seems to rely on both the temperature under which the initial $NO_x$ was generated, as well as any rapid processes taking place once it is emitted into the atmosphere (including photo-chemistry and vertical lofting).

**Table 2.** Distribution of $\alpha_1$ at the corresponding 10[th], 25[th], 50[th], 75[th], and 90[th] percentile values using MFIEF at different industrial sources where CEMS has observations.

| Industrial Sources | $\alpha_1$ | | | | |
|---|---|---|---|---|---|
| | 10[th] | 25[th] | 50[th] | 75[th] | 90[th] |
| Cement | 1.4 | 2.2 | 3.7 | 6.3 | 10.3 |
| Power | 1.4 | 2.0 | 3.3 | 5.9 | 8.6 |
| Iron and Steel | 1.3 | 1.8 | 3.2 | 5.4 | 9.4 |
| Coke | 1.2 | 1.5 | 2.2 | 3.8 | 6.6 |
| Aluminum oxide | 1.3 | 1.4 | 2.6 | 3.5 | 5.0 |
| Boiler | 1.1 | 1.3 | 1.7 | 2.3 | 3.9 |

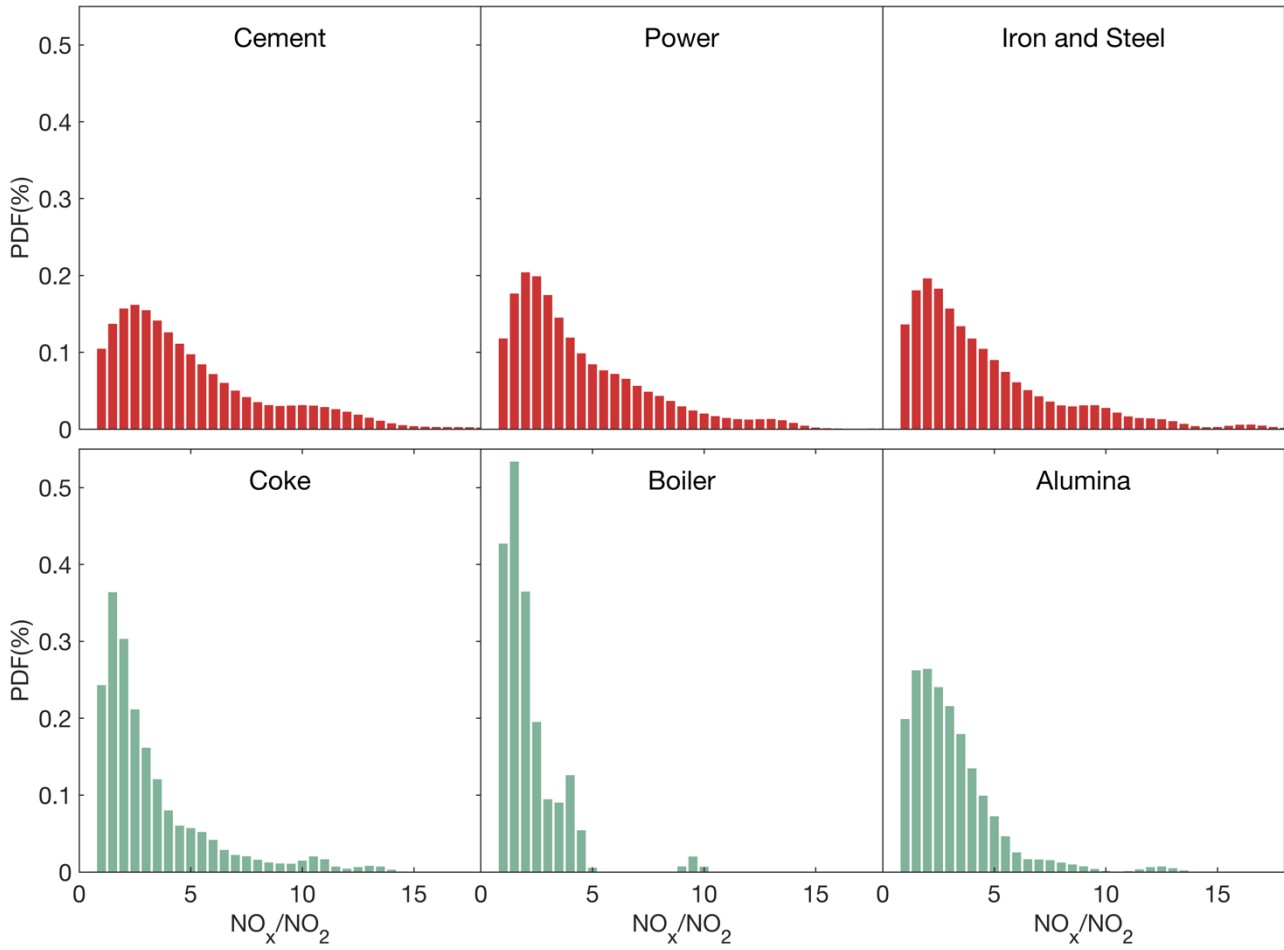

**Figure 14:** Histograms of monthly $\alpha_1$ calculated using MFIEF at the following sources where CEMS has data: cement factories, power plants, iron and steel factories, coke ovens, boilers, and aluminium oxide factories.

## 4 Conclusions

MFIEF based on daily measurements from TROPOMI and a priori daily emissions from CEMS successfully inverts daily $NO_x$ emissions. First, the emissions computed match well with known urban, suburban, and industrial locations. Secondly, the best fit values for thermodynamics ($\alpha_1$) and first order chemical decay ($\alpha_2$) are both physically realistic, while the best fit term for transport ($\alpha_3$) is reasonable based on the mountain and basin geography of the province. Thirdly, the variability of emissions in terms of different geographic location, source types, special events which changed the emissions levels (such as the onset

of COVID-19), and general oxidative, photochemical, and transport conditions of the atmosphere on a monthly scale, are all consistent with what is known. Fourth, the uncertainty is observed to be lower than the day-to-day variability over 30% of the

region, and mostly distributed in regions with higher emission intensities, showing that the results on a day-to-day basis are significant.

MFIEF emissions computed using different a priori datasets (CEMS versus MEIC) yield significant differences from on-the-ground CEMS measurements and local geospatial knowledge of where large sources exist. $EI_{MEIC}$ severely underestimates sources which have a lower amount of emissions, as well as newer sources, while at the same time overestimates sources in Yangquan, Xiaoyi, and enormous steel and power plants, among other similar high emissions sources. $EI_{MEIC}$ incorporates too many low values from MEIC in the suburban and rural areas, leading to many of these grids possessing physically unreasonable or at best very low/high values of $\alpha_1/\alpha_2$ values. A site-by-site comparison with CEMS shows very large differences with $EI_{MEIC}$ at locations which MEIC do have an a priori value, indicating that any a priori dataset must be both precise and accurate, including on a day-to-day basis in order to present a good overall model fit and emissions output product. The MFIEF method as a procedure can quantitatively ally some of these shortcomings of present-day a priori inventories.

The results of variance maximization analysis of 3-years of daily TROPOMI $NO_2$ columns reveal 3 patterns that drive the variability in the $NO_2$ columns. $EI_{CEMS}$ is attributed as responsible for pattern 1, measured UVAI from TROPOMI (which is inversely related to photochemistry) is attributed as responsible for pattern 2, while transport (computed from the gradient of reanalysis wind and TROPOMI $NO_2$ columns) is attributed as responsible for pattern 3. This procedure and its results form a basis of a best-practice approach that the community can adapt to subsequently analyze the efficacy of future emissions products. It is essential to ensure that emissions not only match on average, but also match well with the observed spatial and temporal gradients of the observed remotely sensed fields.

It is observed that the calculated values of $\alpha_1$ are correlated with the thermodynamic conditions of underlying large combustion sources. This offers a new and self-consistent way to quantify the underlying combustion conditions of $NO_x$ generation using remotely sensed measurements. There highest mode and very high values are consistently associated with $\alpha_1$ at cement factories. Generally high modes of $\alpha_1$, with significant values at moderately high and high values are found respectively at iron and steel and power plants. Locations that have CEMS aluminum oxide plants, coke ovens, and boilers are generally lower, but in the case of both aluminum oxide and coke ovens, there is still a significant range of $\alpha_1$ up to 5, while in the case of boilers, there is nearly no high value. There is a slight offset based on the atmospheric temperature, UV radiation, and other atmospheric mixing and chemical forcings, with both colder and lower radiation months having $\alpha_1$ slightly negatively offset and hotter temperature and higher radiation conditions having $\alpha_1$ slightly positively offset. While this offset is consistent across all plant types, it is slightly stronger for the hottest types (cement, iron/steel, and electricity), which are most likely to rise to a higher elevation and therefore be more impacted by the surrounding atmospheric conditions. In all cases, this offset is smaller than the difference between the different source types, indicating that attribution is possible.

The procedure introduced here offers a next step advance in terms of computing emissions from a top-down perspective. Community adaptation and use of these new results will ideally allow improvement in bottom-up inventory constraints and attribution. This work would be improved by reduction in remotely sensed measurement errors/uncertainties, increased use of and access to surface CEMS and other high quality surface flux measurements, improved a priori emission databases, and

higher frequency temporal data availability from new geostationary satellite platforms. The adaptation of day-to-day and other higher frequency quantification data sources, especially sources with well quantified errors would also improve the work herein. The ability to identify large and moderately large industrial sources could be used to identify and quantify sources from many parts of the Global South where ground-based measurements may not be readily available. It is hoped that the findings

herein will be improved upon, possibly in an iterative manner, allowing for more precision and predictability, so that emissions and environmental regulators can have more quantitative support to focus their efforts.

## Data Availability

The satellite $NO_2$ datasets used in this study are available at https://disc.gsfc.nasa.gov/datasets. The ERA-5 reanalysis product is available at https://doi.org/10.24381/cds.bd0915c6. The CEMS online data is available at https://sthjt.shanxi.gov.cn/wryjg.

The MEIC product can be accessed from https://doi.org/10.6084/m9.figshare.c.5214920.v2. All of the data and underlying Figures are available for download at https://doi.org/10.6084/m9.figshare.20459889.

## Author Contributions

Xiaolu Li, Jason Blake Cohen, Kai Qin and Hong Geng developed the research question and set up the whole experimental program. Xiaolu Li wrote the manuscript and performed the data analysis with input from Jason Blake Cohen, Kai Qin, Rui

Zhang, Liling Wu and Liqin Zhang. Liling Wu compared this result with the existing emission inventories and environmental statistics data. Liling Wu and Xiaohui Wu rectify the deviation of CEMS location and make statistics of annual changes. Jason Blake Cohen drafted and corrected the manuscript. Hong Geng, Xiaohui Wu, Chengli Yang, Rui Zhang, and Liqin Zhang supported consultation of the local situation and CEMS data. All authors discussed the results and contributed to the final manuscript.

## Competing Interests

The authors declare that they have no conflict of interests.

## Acknowledgments

The authors would like to thank CEMS for the provision of their emission data. This work thanks the PIs of the TROPOMI, ERA-5, MEIC products for making their data available. The study was supported by the National Natural Science Foundation

of China (42075147), the Shanxi Province Major Science and Technique Program (202101090301013), and the Shanxi Province Postgraduate Education Innovation Program (2021Y035).

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
