# Peer review of "Remotely sensed and surface measurement derived mass-conserving inversion of daily $NO_x$ emissions and inferred combustion technologies in energy rich Northern China"

_EGUsphere, 2023_

## Author Response (AR1)

**Responses (bule) to Reviewer's comments (black)**

Reviewer1

This paper uses a mass balance approach to interpret observations of TROPOMI $NO_2$ columns and CEMS NOx emission flux, enabling the propagation of measured flux at in situ sites to build seamless monthly NOx emission estimates across Shanxi Province, China. It also further interprets the variability of derived model parameters in their framework that represent $NOx/NO_2$ emission ratio (alpha1), NOx lifetime (alpha2) and horizontal advection rate (alpha3). To my knowledge, this is a pioneering study that evaluates and interprets an established space-borne emission estimation method, which has rarely been validated using densely distributed flux observations. The paper is overall well-written with sound methods and results. At the same time, some critical details are missing, and structural changes of the contents are needed. I support the publication of this manuscript, provided that the following comments can be addressed.

Major comments:

1) The MFIEF approach is largely originated from similar box-modeling ideas in previous studies (Beirle et al., 10.1126/sciadv.aax98, 2019; Kong et al., 10.5194/acp-19-12835-2019, 2019), while differs to some extent in details about assumptions on each source/sink process. Line 84-86 presents these studies in the overall "previous study" category, which seem to diminish this connection. I suggest the authors to introduce these approaches in the end of this paragraph, and acknowledge the similarity of idea used in this paper. Also, certain discussions about the uniqueness and capabilities of MFIEF (e.g., using measured emissions, fitting variable parameters that were fixed in previous approaches, etc.) relative to these previous methods should also be added in the Introduction and/or Discussion Section.

We have reorganized the final paragraphs of the Introduction in the following way.

"This approach is partly originated from similar box-modeling ideas in previous studies (Rigby et al., 2008; Beirle et al., 2019; Kong et al., 2019), which themselves are based on previous theory underlying the development of mass-conserving box models (Seigneur et al., 1986). In this specific work, the mass of emissions is connected with

the in-situ observed column loadings through application of the following factors: the temporal rate of change in column loading, first order chemical loss of $NO_x$, gradient transport of $NO_x$, and gradient transport of atmospheric airmass. The coefficients weighting these terms are flexibly fitted, allowing a wider range of possible driving forces and solutions to be considered, while still requiring that these parameters are consistent with observations (Rollins et al., 2012; Karl et al., 2023). The fitted relationship is formed without the use of complex models, can be run on a normal desktop computer, and the end product can be flexibly modified by the user for their own various applications".

The content of other relevant parts has also been changed.

2) The UVAI is used as a proxy of OH and NOx lifetime to interpret EOF2. However, UVAI is a measure of aerosol absorption, while the actual radiation flux reach surface is also sensitive to aerosol scattering, cloud extinction, and solar angles. I did not find existing literature reporting strong correlation between UVAI and OH or NOx lifetime, so it is not convincing for me to justify the interpretation related with Figure 11. Please provide stronger evidence to justify the use of UVAI, or switch to use other parameters (e.g., alpha2?).

We agree that satellite observations of the ultraviolet aerosol index (UVAI) are a measure of aerosol absorption. The absorbing aerosols inferred from the UVAI in turn absorb, scatter, and extinct radiation across all wave bands in the visible and UV portions of the spectrum. This in turn directly impacts all wavebands involved with atmospheric chemistry and climate radiative forcing. This has been demonstrated in numerous studies reporting that absorbing aerosols affect the downwelling surface radiative forcing in the visible (and therefore the actinic flux) (Léon, 2002) as well as OH concentrations (Hammer et al., 2016). Therefore, UVAI indirectly is one component of the chemical decay capacity of $NO_x$ in-situ. Since it is observed on the same platform as the $NO_2$ observations and the reasons provided above, it is introduced here for comparison with EOF2. The fact that they are correlated during peak event times provides strong evidence that during these peak times, the chemical decay of $NO_x$ is strongly related to the in-situ absorbing aerosol column loading.

3) Section 3.3: besides introducing the total uncertainties, contributions from each factor should also be included. One particularly important source is the performance of the fitting and the consequent errors in each parameter. This is the most fundamental information to evaluate the fidelity of the MFIEF framework. How much variability of observed $VNO_2$ and $ENO_x$ can be explained by Eq. (3)? As the fitting is performed including all observations, is it unbiased for all months and grids?

In addition to introducing the total uncertainties, a sensitivity test has been performed to test the robustness of the fits. This is done by making the assumption that the TROPOMI $NO_2$ column values are actually observed near the extreme top and bottom of their $\pm30\%$ uncertainty range. When the TROPOMI $NO_2$ column values are changed, all factors are simultaneously changed. This set of uncertainty runs is applied uniformly as two sperate cases: 70% case (in which the $NO_2$ columns are multiplied by 0.7) and 130% case (in which the $NO_2$ columns are multiplied by 1.3). The coefficients were refit using these new values from TROPOMI and the same values from both CEMS and meteorology over the entire domain included in this work. The results are provided in Response Figure 1.

First, it is observed that in terms of the spatial map and temporal change, that the new $NO_x$ emissions in the 70% case are always larger than 0.7 times the original emissions case (spatially annually averaged and grid-by-grid this varies from 0.89 to 1.00, while temporally domain averaged this varies from 0.77 to 1.04). Similarly, the new $NO_x$ emissions in the 130% case over the median 90% of data is smaller than 1.3 times the original emissions case (spatially annually averaged and grid-by-grid this varies from 1.08 to 1.12 while temporally domain averaged this varies from 1.01 to 1.30).

Second, the constraints on the physically realistic values of $\alpha_1$ and $\alpha_2$, as well as the constant use of CEMS provide a negative feedback loop on the relationship between the $NO_2$ column changes and the final emissions products. This is consistent with the observed computed emissions and differences.

Third, the best fit values of $\alpha_1$, $\alpha_2$, and $\alpha_3$ in each case 70%, 100%, and 130% on a month-by-month basis are always found within the central 50% of the distribution of each other. Furthermore, while the breadth of the different values does change slightly, in some months, it is always either increasing or decreasing on both edges, not only in one direction. These findings indicate that any changes in the parameters between the different $NO_2$ column loading cases are generally smooth, consistent, provide redundancy to each other, and are also influenced significantly by the a priori emissions used in the fitting.

[Figure]

[Figure]

**Response Figure 1**. Different TROPOMI NO$_2$ for emission calculation based on CEMS using MFIEF: (a) 70%, 100%, 130% cases calculated emission and their uncertainty; (b) the differences between 70% to 100% case and 130% to 100% case; (c) time series of 70%, 100%, 130% cases and their differences; (d) $\alpha_1$, $\alpha_2$, and $\alpha_3$ in each case.

4) Section 3.4: The derived emissions are representative of ambient fluxes (instead of initial emissions from the furnace), so the rapid NO-NO$_2$ conversion and consequently the NOx/NO$_2$ ratio is dependent on not only the combustion environment factors discussed in the manuscript, but also ambient chemistry (e.g., photolysis rate and ozone concentration). I assume the latter factor might be more important in driving the seasonal variations in Figs. 12 and 14. Due to the lack of full consideration of all driving factors as well as the lack of outstanding hotspot of alpha1 from certain month or factory, the current discussion of alpha1 in this section is relatively more conjectural than the other part of the paper. My overall suggestion is to greatly reduce the amount of discussion and focus on 1-2 most convincing observations that can be concluded from existing data, with acknowledgement of various factors driving alpha1 variability and precluding a full explanation of all revealed variabilities.

Initially, the ratio of $\alpha_1$ is entirely determined at the source as a function of the type of source, its thermodynamic conditions, availability of nitrogen and oxygen, water vapor, etc. However, after emissions into the atmosphere there is a rapid adjustment that occurs from the extremely hot air emitted at the stack or pipe exit until it comes to equilibrium in terms of both vertical height and thermodynamic condition. During this period of time, further modification occurs.

However, the assertion that such chemistry or thermodynamic changes are important is based on many assumptions, which themselves are not necessarily valid in the real atmosphere.

1. There are many observations in this area that demonstrate at times there is insufficient ozone present at the surface to convert NO$_2$ to NO even on days which have a high surface temperature, which should be the times with the largest surface O$_3$ concentration (Response Figure 2).
2. The column ozone is even lower than the surface ozone, so the fraction of the emissions which buoyantly break into the free troposphere or are lofted by upslope winds will always encounter ozone which is too low to lead to chemical titration.

[Figure]

[Figure]

**Response Figure 2**. Time series of hourly concentration of $O_3$ and $NO_2$ from ambient air quality monitoring stations near an iron and steel factory in Taiyuan City.

Furthermore, while there is modification observed from different months of the year on the mean and standard deviation of the value of $\alpha_1$, that the types with larger $\alpha_1$ are still larger and the types with smaller $\alpha_1$ are still smaller (Response Figure 3). This indicates that indeed the original thermodynamics still plays an important role in determining the value of $\alpha_1$.

Since the climatology across the regions which have these sites is roughly similar, the only things over the 3-years of data that can distinguish these sites from each other are the source type of emissions and the TROPOMI $NO_2$ column loading (Response Figure 4).

[Figure]

**Response Figure 3.** Distribution of $\alpha_1$ during certain warm and cold month at cement factories, power plants, steel and iron factories, coke ovens, boilers, and aluminium oxide factories.

[Figure]

**Response Figure 4.** Histogram of $\alpha_1$ calculated based on CEMS using MFIEF at cement factories, power plants, steel and iron factories, coke ovens, boilers, and aluminum oxide factories.

Specific comments:

1) Line 22-23: As outlined before, the statement of "significant correlation with combustion temperature and energy efficiency" might be too strong here.

We have slightly toned down the strength of this statement.

2) Line 39: delete "are more serious".

We agree and have removed this phrase.

3) Line 48: delete "(2015, 2020)".

The reference format has been modified.

4) Citations in the Introduction Section:
Line 42: should also cite Zhang et al., 10.1073/pnas.1907956116, 2020; Wang et al., 10.1073/pnas.2007513117, 2020; Li et al., 10.1016/j.scitotenv.2021.150011, 2022; Wei et al., 10.5194/acp-23-1511-2023, 2023.

These (and other) references have been added to the revised version. Thank you for pointing out these interesting scientific works.

Line 54: (Beirle et al., 2011) is not a paper studying NOx forming aerosol.

We agree. We believe that Rollins et al. (2012) is a better fit here. This has been updated.

Line 56: Besides China, some other regional inventories (e.g., McDonald et al., 10.1021/es401034z, 2013; Xing et al., 10.5194/acp-13-7531-2013, 2013) might be worth citing.

These (and other) references have been added to the revised version. Thank you for pointing out these interesting scientific works.

Line 92: can cite (Zheng et al., 10.5194/acp-18-14095-2018, 2018) for MEIC.

This line has been moved to the Materials and Methods section and the reference has been appropriately updated.

5) Line 85-86: As outlined before, should clarify that your approach improves these assumptions to some extent.

We have deleted this sentence, and further reorganized the entire final paragraph of the Introduction. It now reads:

"This approach is partly originated from similar box-modeling ideas in previous studies (Rigby et al., 2008; Beirle et al., 2019; Kong et al., 2019), which themselves are based on previous theory underlying the development of mass-conserving box models (Seigneur et al., 1986). In this specific work, the mass of emissions is connected with the in-situ observed column loadings through application of the following factors: the temporal rate of change in column loading, first order chemical loss of $NO_x$, gradient transport of $NO_x$, and gradient transport of atmospheric airmass. The coefficients weighting these terms are flexibly fitted, allowing a wider range of possible driving forces and solutions to be considered, while still requiring that these parameters are consistent with observations (Rollins et al., 2012; Karl et al., 2023). The fitted relationship is formed without the use of complex models, can be run on a normal desktop computer, and the end product can be flexibly modified by the user for their own various applications." Showing some extent of us on the box-modeling and mass balance idea.

6) Line 91-92: These two are bottom-up inventories, so should follow after Line 73? I do not see clear connection of this sentence with the previous text.

This sentence has been modified and moved to the Materials and Methods section.

7) Line 98: What idea from bottom-up inventory is used in your approach?

The MFIEF approach uses an emissions inventory as the a priori to begin the inversion. This work uses two different emissions inventories: one is bottom-up derived from observed CEMS fluxes, while the other is fully bottom up MEIC.

8) Line 107: As outlined before, using UVAI as a proxy of UV radiation seems not appropriate.

As we have responded above, UVAI is a measurement that provides information on the column loading of absorbing aerosol in the UV. This absorbing aerosol in turn impacts the radiative flux at all visible and UV bands through absorption, scattering, and extinction. There has been extensive work which has demonstrated that the absorbing aerosols reduce the actinic flux and alter OH. In both cases, this has an impact on the atmospheric lifetime of $NO_x$. We agree that this is not the only component, but we do believe that it is fair to say that UVAI has an impact on the net actinic flux at the surface.

We have modified the paper as: "different actinic flux and atmospheric oxidation"

9) Fig. 3c and 4c: set log-scale for x-axis might increase the readability of the figure. Also, 28% of days are absent in 2019 so would that affect the sampling and representativeness of data in Fig. 3?

The x-axis has been changed into log-scale (Response Figure 5).

Since there is insufficient CEMS data for November and December 2019 to fit the equations, the results of the month-by-month calculations of emissions during those two months is reflective of the fitted values of $\alpha_1$, $\alpha_2$, and $\alpha_3$ from other similar conditions, which includes data from November and December in other years, as well as surrounding data from January and February. These are conditions in which there should be somewhat similar climatological factors such as temperature, actinic flux, wind, and other environmental data which impacts upon the observed $NO_2$ column from TROPOMI. At all times, the actual TROPOMI $NO_2$ column observations are used to constrain its emissions field. This is consistent with the production methods of

the companies and the requirements for production stoppages and restrictions in general.

[Figure]

**Response Figure 5.** PDFs of day-by-day and grid-by-grid emissions of CEMS and MEIC over individual years (with log-scale for x-axis).

The CEMS data from 2018 through the present indicates that the missing times in 2019 are found near the median range of the distribution, and therefore are not biased.

10) Line 211: this is true for daytime and locations with strong $NO_x$ emissions only. See (Kenagy et al., 10.1029/2018JD028736, 2018) for nighttime sinks, and (Romer Present et al., 10.5194/acp-20-267-2020, 2020) for possible significant daytime sink via reactions with $RO_2$. As Eq. 3 relates $VNO_2$ at afternoon overpass to 24-h mean emissions, the lifetime should also reflect all hours during the day.

We agree that this work is reflective of the different chemical loss sources which occur throughout the day, and also throughout the column where the emissions spread to. This must therefore include actinic flux derived chemical reactions (i.e., $RO_2$ during the daytime), heterogenous surfaces (i.e., $N_2O_5$), and other reactions which

happen in the free troposphere under much colder and lower pressure conditions as a considerable amount of the flux is rapidly brought up to elevation by upslope winds in this region. The net linear coefficient $\alpha_2$ is a reflection of the net total 24-hour, atmospheric column chemical first order loss coefficient. This is a very interesting area for further study to see if and how simple non-linearity could be brought into better constraining the chemical loss term for future applications of this work.

The following changes have been made: "The second of these is the chemical loss of $NO_x$, which will always lead to a decrease in the stock. The chemical sink of $NO_x$ is dominated by the reaction between $NO_2$ and OH, via reactions with products formed from the actinic flux (i.e., chemistry such as $RO_2$), and on aerosol surfaces via heterogeneous reactions (Valin et al., 2013; Kenagy et al., 2018; Romer Present et al., 2020), which herein is described as S".

11) Equation 3: Since $VNO_2$ is a snapshot of afternoon overpass while ENOx is 24-h average, so alpha1-alpha3 all contain the conversion from overpass time to 24-h mean. Should acknowledge this fact.

This part is now explained in greater detail, including the fact that TROPOMI has some days with a single overpass, and other days with two separate overpasses approximately 101.5 minutes apart at the same location. During this specific subset of days, information from both overpasses is used on average.

"$V_{NO2}$ is observed as either one or two overlapping snapshots of total column information occurring at 13:30 LT (and under some conditions also 101.5 minutes either earlier or later (Tonion and Pirotti, 2022)). In all cases, the meteorological values and CEMS values are representative of 24-hour total and/or daily average conditions respectively. Therefore, the fitted values of $\alpha_1$, $\alpha_2$, and $\alpha_3$, as presented are representative of 24-hour average or 24-hour net effect respectively, acting on the entire column of $NO_x$".

12) Fig. 5: As outlined before, alpha1 is not just determined by type of source.

$\alpha_1$ is determined in significant part by the type of source, its thermodynamic conditions, and availability of nitrogen and oxygen. This is also modified based on rapid chemistry or thermodynamics which occur in the in-situ atmosphere. However, there are many observations in this area which demonstrate that at times there is insufficient ozone present to convert $NO_2$ to NO, indicating that in many cases, the rapid atmospheric adjustment is not actually happening. Furthermore, the emissions include not only what ends up in boundary layer, but also the fraction above the boundary layer, which occurs through upslope winds and plume rise. The chemistry above the boundary layer tends to be far slower and the controlling factors frequently are different in nature. The effects of the change also are averaged over 24 hours, and therefore include night-time as well as day-time types of effects, as previously mentioned.

13) Fig. 6: a scatter plot of Fig. 6a vs. Fig. 3a will provide an insight about how representative Eq. 3 is. Certain locations with strong emissions while unmeasured by Fig. 3a should also be discussed (e.g., are these exactly locations of missed stationary sources?).

PDFs of the a priori emissions (CEMS, Fig. 3a), $EI_{CEMS}$ (Fig. 6a) at locations which have CEMS data, and $EI_{CEMS}$ at locations which do not have CEMS data, are calculated PDFs using all data on a day-by-day and grid-by-grid basis in Response Figure 6. As demonstrated, the value of emissions computed at CEMS sites is slightly larger in the mean and median than the values of emissions computed off CEMS sites, in particular between emissions in the range from 0.5 to 1.5 $\mu g\ m^{-2}\ s^{-1}$. However, at the extremes even the $EI_{CEMS}$ locations without CEMS data has both high and very low values. This is due to the fact that CEMS sites do not include traffic, residential, and small industrial sources. Therefore, there are some net high emissions sources in some regions that have no CEMS data available.

These results demonstrate that this approach is sufficiently flexible that it can be applied to identify and roughly estimate emissions from areas in which the conditions are similar to but not absolutely the same as those at which the training occurs. We have demonstrated that there is enough data from the existing CEMS network to train the

model to reproduce emissions over the entire range of values observed in Shanxi. This is due to the fact that the training has occurred over a sufficiently wide range of input conditions, TROPOMI $NO_2$ columns, meteorology, and other forcing factors.

The comparison with the actual CEMS data shows that both emissions datasets are more central than the CEMS data. This also makes sense, since there is no grid in which 100% of the total sources are due to only CEMS. It also is factually true that in the real atmosphere, the actions of transport and diffusion will tend to reduce very large values and fill in very small values.

[Figure]

**Response Figure 6.** The distribution of $EI_{CEMS}$ with and without CEMS data, and CEMS.

14) Fig. 8: What spatial extent is used to calculate the city-mean emissions? If the range is too small, the difference between MEIC and CEMS could be dominated by dilution by the large grid cell.

First, our resolution is 0.05°×0.05°, which is higher than the 0.25°×0.25° MEIC product used. However, the number of grids per city is far more than the offset ratio of 25. The

number of grids in each city is show on the table below. We think that the grids number is enough to calculate the difference between MEIC and CEMS.

**Response Table 1**. The number of grids in each city

| City | TY | DT | YQ | CZ | JC | SZ | JZ | YC | XZ | LF | LL |
|------|-----|-----|-----|-----|-----|-----|-----|-----|------|-----|-----|
| No. of grids | 289 | 591 | 183 | 566 | 375 | 446 | 663 | 565 | 1045 | 818 | 866 |

15) Fig. 9: Are the spatial distribution of EOF1 correlating with that of alpha1? How about EOF2 vs. alpha2? EOF3 vs. divergence?

Spatial correlation was performed grid-by-grid between the three-year average of $\alpha_1$ and EOF1, where r=0.18, p<0.01, indicating that there is a statistically significant correlation, but one which is far less significant than the result currently presented. Similarly, correlation was performed grid-by-grid between three-year average of $\alpha_2$ and EOF2, where r=0.15, p=0.012, indicating that there is also a statistically significant correlation, but one which is far less significant than the result already presented. The correlation between EOF3 and divergence is already displayed in the paper (fig.12).

16) Sections 3.1 and 3.2: Alpha1-alpha3 all exhibit certain spatial and temporal variabilities. What are the implications on previous methods that have simpler (e.g., fixed) assumptions?

In Beirle et al. (2019) $\alpha_1$=1.32 ± 0.26 and $\alpha_2$=4 ± 1.3 hours. These results determine that 19% of total sites have as value of $\alpha_1$ inside of their range of fixed $\alpha_1$, while 79% of sites have a value of $\alpha_1$ larger and 2% of sites have a value of $\alpha_1$ smaller than allowed by previous approaches. Furthermore, only 4% of the total sites in this work have as value of $\alpha_2$ inside their range of fixed $\alpha_2$ and 96% of sites have a value of $\alpha_2$ larger than allowed by previous fixed assumption approaches. For these reasons, using the fixed assumptions approach would lead to a large majority of the grids in this work having an emissions value which is not properly predicted. The magnitude of emissions is also biased based on their range, in particular with Power Plants/Steel Factories/Cement factories having values of $\alpha_1$ and $\alpha_2$ which are far outside of the

ranges of their fixed studies, while also being grids with higher absolute emissions values.

17) Line 462: should mention possible benefits from Geostationary instruments that can be promising to resolve the expectedly strong diurnal variability of alpha1-alpha3.

This is an excellent suggestion. The following change has been made to the text:

This work would be improved by reduction in remotely sensed measurement errors/uncertainties, increased use of and access to surface CEMS and other high quality surface flux measurements, improved a priori emissions databases, and higher frequency temporal data availability from new geostationary satellite platforms.

**Reviewer3**

**General comments:**

This study presents a new model-free method to constrain NOx emissions using TROPOMI $NO_2$ and ERA-5 wind data. The new method is based on mass balance theory and considers the $NO_x/NO_2$ ratio, $NO_x$ lifetime and $NO_x$ transport. Based on this approach, daily $NO_x$ emissions over Shanxi province are estimated during 2019-2021. Some comments should be addressed before its publication. Additionally, the authors apply EOF to TROPOMI $NO_2$ and relate the first three PCs to $NO_x$ emissions, UVAI, and $NO_x$ transport. The method and conclusions are important, but some comments should be addressed before its publication.

**Specific comments:**

1.  When TROPOMI $NO_2$ is not available due to cloud or other reasons, how do you deal with it? My understanding is when TROPOMI $NO_2$ is not available, Eq. 3 does not work. And how the missing data affect the estimated emission inventory?

This paper introduces a new methodology and makes a first attempt on the combine use of $NO_2$ column loadings and high spatial and temporal frequency observations of ground emissions, within the confines of a first order approximation to the overall mass balance framework. You are correct in that when and where there is missing data, that the emissions cannot be calculated at that exact place and time. We have

examined the PDFs of the output emissions at each location and found that they are relatively smooth. For these reasons, unless the missing $NO_2$ observation were statistically very high or very low compared with the other values that already exist, they would not make a large difference in the overall emissions. However, including more observations from other existing and new observation platforms, or using other remotely sensed species in tandem will also help to improve the emissions estimate. Thank you for this suggestion, as it provides a path for future work.

2. In Sect 3.1, why a priori emission inventory (CEMS or MEIC) is needed to estimate new emission inventory? According to Eq. 3, it does not require a priori emission inventory.

In order to fit the first order terms approximating thermodynamics $\alpha_1$, chemistry $\alpha_2$, and transport $\alpha_3$ in Eq. 3, an initial guess of emissions is required to complete the multiple linear regression. This then allows the distributions of the parameters $\alpha_1$, $\alpha_2$, and $\alpha_3$ to be subsequently used in Eq. 3 to calculate the final emissions. It also allows for error analysis, since the fitted terms themselves have a range of possible solutions. In this work, two different emissions a priori were selected, with the goal being to demonstrate what differences this would have on the computed emissions.

This procedure is similar to how chemical transport models (including GEOS-Chem, WRF-Chem, etc.) have their initial uncertain variables fitted. The major differences being in this work the variables are sufficient simple so as to be flexible and presented in an open way. This allows for a wider range of possible emissions datasets to work within the model environment, which may not be possible with more heavily fitted or constrained modeling approaches. We believe the work herein demonstrates robustness as an entire system.

3. How EOF is applied to the daily TROPOMI $NO_2$ columns when data are not available in some grids?

When TROPOMI $NO_2$ columns are not available in some grids, the climatological average value in that grid is assigned in order to compute the EOF. The grid is also tagged and after the EOF is computed, the grid in space and time is reset to NaN, following Cohen (2014).

4. I'm curious why the seasonal variation of $NO_x$ lifetime shown in Fig. 13b is so small. For example, Lamsal et al. (2010) estimated that the lifetime is $NO_x$ is 7.6 h in summer and 17.8 h in winter, while this study showed that lifetime is ~12 h regardless of season.

Lamsal, L. N., Martin, R. V., van Donkelaar, A., Celarier, E. A., Bucsela, E. J., Boersma, K. F., Dirksen, R., Luo, C., & Wang, Y. (2010). Indirect validation of tropospheric nitrogen dioxide retrieved from the OMI satellite instrument: Insight into the seasonal variation of nitrogen oxides at northern midlatitudes. Journal of Geophysical Research: Atmospheres, 115(D5). https://doi.org/10.1029/2009JD013351

The median values of $NO_x$ lifetime do demonstrate a range from 9.0 hours to 14.7 hours in different months. The 10th and 90th percentile values match with your reference paper quite well, being 7.1 hours and 18.1 hours respectively. The results in this work are based on the total column values, which includes temperature, UV, climate, and aerosols which are observed in Shanxi. Based on the results herein, the largest values are found in June or July and the smallest values are found in September. This is due to the complex local conditions and forcing factors including the complex boundary layer height, the variable aerosol loading, cloudiness, and other factors.

5. The authors concluded that "Thirdly, the general variability in geography, month of the year, and years before and after COVID-19 are all consistent with what is known.", while readers cannot find any analysis that is related to COVID-19 in the manuscript.

The results herein show clearly that the emissions before COVID-19 were higher than after COVID-19. In specific, the time series shows that while there is a variation as a function of the time of the year, there is also a disturbance in this variation due to the timing of onset of COVID-19. What is important is that the emissions results match well with what is known by the community in terms of month-to-month changes, and geographic diversity, variability, and consistency across different industrial sources and under different oxidative and transport conditions. We have reorganized here in the following way:

"Thirdly, the variability of emissions in terms of different geographic location, source types, special events which changed the emissions levels (such as the onset of COVID-19), and general oxidative, photochemical, and transport conditions of the atmosphere on a monthly-scale, are all consistent with what is known."

**Technical corrections:**

Line 202: the first three PC account for less than 50%

The first point is that the community acknowledges there is an uncertainty in TROPOMI observations of $NO_2$ which ranges as high as 30% to 50%. In this case, 30% to 50% of the PCs will be representative of this uncertainty, meaning their signal pattern while mathematically correct, is physically meaningless. Therefore, the results herein represent nearly all of the remaining variability. The three spatial modes [EOF1, EOF2, and EOF3] contribute 29.4%, 8.4%, and 4.4% respectively (accounting for 42.2%), while the fourth mode onward all contribute less than 4.0%. Next, the first three modes all have a high degree of correlation with known underlying driving phenomenon, while the fourth mode and onward show no such relationship. There are many other factors that affect the pollutant column loading in the atmosphere, and if we had more data to analyze, a way to bring in more variables, or a way to reduce the uncertainty, we would also like to search for and work more on attribution. Thank you for helping to carefully guide and clarify our thought process.

Line 26: $NO_2$ columns identifies -> $NO_2$ columns, which facilitates to identify

Thank you, it has been modified.

Line 47: However -> Moreover

Thank you, it has been modified.

Line 49: also impacting -> that impact

Thank you, it has been modified.

Line 51: Nitrogen Monoxide -> nitric oxide

Thank you, it has been modified.

Line 60: statistics on -> statistics for

Thank you, it has been modified to "statistics representing".

Line 67: differences -> the differences
Thank you, it has been modified.

Line 84-85: Kong et al. (Kong et al., 2019) and Beirle et al. 85 (Beirle et al., 2019) ->
Kong et al. (2019) and Beirle et al. (2019)
Thank you, it has been modified.

Line 88: costly -> cost
Thank you, "costly to run" has been modified to "computationally intensive".

Line 106-109: "The fact that …… variations observed." This sentence is not easy to
understand; it is better to rewrite.
We have reorganized here in the following way.
"This method has been used in different situation such as over different months, over
multi-year changes in the environment, under different actinic flux and atmospheric
oxidation conditions, under complex meteorological domains, and over sources which
are both thermodynamically stable as well as unstable. That permits this study to
explore the full range of variations".

Line 126: Henk Eskes, 2021 -> Henk et al., 2021
The reference format has been modified.

Line 126-128: "Furthermore, …… being discarded." This sentence is not easy to
understand; it is better to rewrite.
We have reorganized here in the following way.
"Furthermore, an additional filter is applied to set all individual gird of $NO_2$ column
which is less than $1.4\times10^{15}$ molec cm$^{-2}$ to be NaN. This is done to avoid issues where
the observed signal may be smaller than the uncertainty of the signal itself (J.H.G.M
Van Geffen, 2021; Qin et al, 2022)".

Line 130: 2021are - > 2021 are
Thank you, it has been modified.

Line 149: as discharged -> emitted

Thank you, it has been modified.

Line 151: NOx concentration measuring -> measuring NOx concentrations

Thank you, it has been modified.

Line 163: 24 is convert -> 24 is used to

Thank you, it has been modified to "24 is used to convert units from hours to days".

Line 168: highest -> the highest

Thank you, it has been modified.

Line 179: uniformity -> uniformly

Thank you, it has been modified.

Line 201: (Björnsson and Venegas, 1997) and (Cohen, 2014) -> Björnsson and Venegas (1997) and Cohen (2014)

Thank you, it has been modified.

Line 208: transport to -> transport

Thank you, it has been modified.

Line 222: basis. -> basis

Thank you, it has been modified.

Line 225: and not -> rather than

Thank you, it has been modified.

Line 232: $\alpha 1$ $\alpha 2$ and $\alpha 3$ -> $\alpha 1$, $\alpha 2$, and $\alpha 3$

Thank you, it has been modified in the whole paper.

Line 259: area. -> area,

Thank you, it has been modified.

Line 387: Fig. 12(a) -> Fig. 12a

Thank you, it has been modified in the whole paper.

**References**

Beirle, S., Borger, C., Dorner, S., Li, A., Hu, Z. K., Liu, F., Wang, Y., and Wagner, T.: Pinpointing nitrogen oxide emissions from space, Sci. Adv., 5, https://doi.org/10.1126/sciadv.aax9800, 2019.

Cohen, J. B.: Quantifying the occurrence and magnitude of the Southeast Asian fire climatology, Environ. Res. Lett., 9, https://doi.org/10.1088/1748-9326/9/11/114018, 2014.

Hammer, M. S., Martin, R. V., van Donkelaar, A., Buchard, V., Torres, O., Ridley, D. A., and Spurr, R. J. D.: Interpreting the ultraviolet aerosol index observed with the OMI satellite instrument to understand absorption by organic aerosols: implications for atmospheric oxidation and direct radiative effects, Atmos. Chem. Phys., 16, 2507-2523, https://doi.org/10.5194/acp-16-2507-2016, 2016.

J.H.G.M. van Geffen, H. J. Eskes, K.F. Boersma and J.P. Veefkind: TROPOMI ATBD of the total and tropospheric $NO_2$ data products, Open File Rep., 2021.

Karl, T., Lamprecht, C., Graus, M., Cede, A., Tiefengraber, M., Vila-Guerau de Arellano, J., Gurarie, D., and Lenschow, D.: High urban $NO_x$ triggers a substantial chemical downward flux of ozone, Sci. Adv., 9, eadd2365, https://doi.org/doi:10.1126/sciadv.add2365, 2023.

Kenagy, H. S., Sparks, T. L., Ebben, C. J., Wooldrige, P. J., Lopez-Hilfiker, F. D., Lee, B. H., Thornton, J. A., McDuffie, E. E., Fibiger, D. L., Brown, S. S., Montzka, D. D., Weinheimer, A. J., Schroder, J. C., Campuzano‐Jost, P., Day, D. A., Jimenez, J. L., Dibb, J. E., Campos, T., Shah, V., Jaeglé, L., and Cohen, R. C.: $NO_x$ lifetime and $NO_y$ partitioning during winter, J. Geophys. Res. Atmos., 123, 9813-9827, https://doi.org/10.1029/2018jd028736, 2018.

Kong, H., Lin, J., Zhang, R., Liu, M., Weng, H., Ni, R., Chen, L., Wang, J., Yan, Y., and Zhang, Q.: High-resolution ($0.05° \times 0.05°$) $NO_x$ emissions in the Yangtze River Delta inferred from OMI, Atmos. Chem. Phys., 19, 12835-12856, https://doi.org/10.5194/acp-19-12835-2019, 2019.

Léon, J. F.: Aerosol direct radiative impact over the INDOEX area based on passive and active remote sensing, J. Geophys. Res., 107, https://doi.org/10.1029/2000jd000116, 2002.

Qin, K., Shi, J., He, Q., Deng, W., Wang, S., Liu, J., and Cohen, J. B.: New Model-Free Daily Inversion of $NO_x$ Emissions using TROPOMI (MCMFE-$NO_x$): Deducing a See-Saw of Halved Well Regulated Sources and Doubled New Sources, ESS Open Archive [preprint], https://doi.org/10.1002/essoar.10512010.1, July 26, 2022.

Rigby, M., Prinn, R. G., Fraser, P. J., Simmonds, P. G., Langenfelds, R. L., Huang, J., Cunnold, D. M., Steele, L. P., Krummel, P. B., Weiss, R. F., O'Doherty, S., Salameh, P. K., Wang, H.

J., Harth, C. M., Mühle, J., and Porter, L. W.: Renewed growth of atmospheric methane, Geophys. Res. Lett., 35, https://doi.org/10.1029/2008gl036037, 2008.

Rollins, A. W., Browne, E. C., Min, K. E., Pusede, S. E., Wooldridge, P. J., Gentner, D. R., Goldstein, A. H., Liu, S., Day, D. A., Russell, L. M., and Cohen, R. C.: Evidence for $NO_x$ Control over Nighttime SOA Formation, Science, 337, 1210-1212, https://doi.org/10.1126/science.1221520, 2012.

Romer Present, P. S., Zare, A., and Cohen, R. C.: The changing role of organic nitrates in the removal and transport of $NO_x$, Atmos. Chem. Phys., 20, 267-279, https://doi.org/10.5194/acp-20-267-2020, 2020.

Seigneur, C., A. B. Hudischewskyj, J. H. Seinfeld, K. T. Whitby, E. R. Whitby, J. R. Brock, and H. M. Barnes, Simulation of aerosol dynamics: A comparative review of mathematical models, Aerosol Sci. Technol., 5, 205-222, https://doi.org/10.1080/02786828608959088, 1986.

Tonion, F. and Pirotti, F.: Sentinel-5p $NO_2$ data: cross-validation and comparison with ground measurements, ISPRS Archives, XLIII-B3-2022, 749-756, https://doi.org/10.5194/isprs-archives-XLIII-B3-2022-749-2022, 2022.

Valin, L. C., Russell, A. R., and Cohen, R. C.: Variations of OH radical in an urban plume inferred from $NO_2$ column measurements, Geophys. Res. Lett., 40, 1856-1860, https://doi.org/10.1002/grl.50267, 2013.

---

## Author Response (AR2)

**Responses (bule) to Reviewer's comments (black)**

The authors addressed most of the two reviewers' comments adequately. Meanwhile I still have reservations about two remaining issues as follows. I support the publication of this paper if they can be further addressed.

1. I agree that UVAI can be used as a proxy of aerosol absorption and OH abundance to some extent, but please further explain 1) why is the absolute value of UVAI used in the analysis (Figure 11, negative UVAI implies aerosol scattering, not absorption); 2) should some absolute abundance of aerosol also be used (e.g., AOD) since UVAI is more of an indicator of relative absorption vs. scattering?

Large negative values of UVAI (especially more negative than -0.5) correspond to high AOD values (larger than 0.5) and concurrent SSA values generally ranging from 0.92 to 0.98, as observed at multiple AERONET stations (Figs 1, 6, 8 from Penning et al., 2009). While these values are not as absorbing as large positive values of UVAI (especially more positive than 1.0) which correspond to SSA values less than 0.9 (Torres et al., 2020), both are still absorbing due to the high AOD values present (1.0 to 3.0 or possibly even more). Analyzing all of the data from 2019 through 2021 on a grid-by-grid basis, there are found to be 357171 grids (5.0%) with positive values and 6740079 (94.4%) grids with negative values. The negative UVAI values observed over Shanxi are found to be very negative, as shown in the results before, consistent with and even stronger than the AERONET conditions reported by Penning et al. (2009).

There is a statistically significant correlation between the negative UVAI and EOF2 in both the special and temporal dimensions, although it is slightly smaller than the correlation shown in Figure 11 of the paper, and of the opposite sign, since the majority of values of UVAI which are far from zero are negative. The PDFs of the UVAI values over each of the subregions as EOF2 is increased are also recalculated using the negative UVAI values only, and demonstrate that the results are again nearly the same as in the original Fig.11 in the paper, but found to just be of the opposite sign.

We conclude that in this case, the use of the absolute value of UVAI is most appropriate and reasonable, since given the relatively high AOD levels found on average, even particles with a small amount of absorption still will have a significant

impact on the UV radiation. For these reasons, the analysis using the absolute value of UVAI was explained more in depth, and the following changes have been made in the paper:

Second, it is asserted that EOF2 is related to TROPOMI measured UVAI, which physically makes sense, since satellite observations of UVAI are sensitive to aerosol extinction in the UV, with very large values indicating large amounts of highly absorbing aerosol (SSA less than 0.9) and very large negative values indicating large amounts of partially absorbing aerosol (SSA between 0.92 and 0.98) (Penning et al., 2009; Torres et al, 2020). There have been numerous studies reporting that absorbing aerosols affect the downwelling surface radiative forcing in the visible (and therefore the actinic flux) (Léon, 2002) as well as OH concentrations (Hammer et al., 2016). Therefore, UVAI is an indirect proxy of an aspect contributing to the chemical decay capacity of $NO_x$ in-situ. Applying four different cutoffs to EOF2, it is observed that as the EOF2 domain increases in magnitude, that the 3-year mean measured absolute value of TROPOMI UVAI becomes smaller in magnitude , as demonstrated in Fig. 11. Since values of UVAI closer to zero imply less atmospheric extinction (absorption for positive values and a mixture of scattering and absorption for negative values), it therefore also scales inversely with surface UV radiation, implying that when the UVAI is lower, that there is more available UV radiation, and hence implicitly faster chemical decay of $NO_x$. This is consistent with the UV radiation being responsible for the second mode of the maximized variance. The negative correlation observed between absolute values of UVAI weighted by high absolute values of EOF2 (EOF2>0.02) grid-by-grid in the same EOF2 region is anticorrelated with PC2 ($r$=-0.33, p<0.01). While $r$ in this case is a lot smaller, it is also consistent with theory, since in order to significantly affect the OH levels, the changes in UV radiation and hence UVAI must be very large, which is found to not occur frequently in time, but when it does occur, it makes a significant impact.

[Figure]

**2nd Response Figure 1 (new Figure 11):** Four different cutoffs of EOF2 are used to set the spatial domains. The maps in (a-d) are plots of EOF2/|UVAI| where the cutoffs are given as (a) EOF2 >0.005, (b) EOF2 >0.01, (c) EOF2 >0.015, (d) EOF2 >0.02. (e) Histograms of |UVAI| over the domains given respectively in a-d. (f) Time series of weekly PC2 and |UVAI| in the (d) domain.

Your suggestion to use an absolute abundance of aerosol (e.g., AOD or AAOD) would provide an interesting point for future work. This is in our planned set of future work, as we have already used AAOD in previous works elsewhere in the past (i.e., Cohen and Wang, 2014; Wang et al., 2021), although at the present time this is not easy to do over Shanxi, since there are no AERONET stations available over this region for validation. There is a Chinese SONET station which was recently installed, and perhaps it can provide the validation required for this as future work.

2. I still disagree that the alpha1 can really be used to distinguish the source categories. 1) Response Figure 2: the anti-correlation between $O_3$ and $NO_2$ exactly suggests strong conversion from NO to $NO_2$ by $O_3$. I do not understand the way the authors used this example to indicate insufficient conversion?

There are a few points that we use to guide our response to this interesting point. First of all, surface observations clearly demonstrate that there are frequently observations found where there is little to no ozone at the surface. Therefore, during these times of day, the conversion from NO to $NO_2$ by $O_3$ will not occur or will occur so slowly as to not impact the ratio of the emissions occurring at these times of the day. Hence, during this fraction of the day, the ratio of emissions is almost 100% controlled by the thermodynamic relationship from $\alpha_1$. This is particularly so for emissions from large sources observed by CEMS, which tend to run roughly evenly throughout the 24-hour day. Second, there is an even longer period of time where the concentration of ozone is not zero, but is still much lower than $NO_2$, indicating that even if it is exerting some control on the NO to $NO_2$ conversion, that it is not dominant when compared with the thermodynamic relationship from $\alpha_1$ on freshly emitted $NO_x$ during these times of the day. Third, due to the buoyancy from the heat co-emitted with the $NO_x$ from these large sources, a significant amount of these fresh emissions will be lofted into the free troposphere, wherein the ozone values are even lower than at the surface, and therefore the impacts will also be less.

In order to quantify the relationship between $O_3$ and $NO_2$, we have conducted a one-year long (2019) analysis of surface observations in Taiyuan City, which is both in the domain of this work, and located near one of China's largest individual iron and steel sources. We use $5\mu g/m^3$ and $25\mu g/m^3$ of $O_3$ concentration as the dividing line to separate almost no ozone available to titrate $NO_x$ and as there being some ozone but insufficient to fully titrate to chemical equilibrium over the hour-long time period of the observations. There are four months in which every day has more than 3 hours that the $O_3$ concentration smaller than or equal to $5\mu g/m^3$ and four months with at least half of the days under similar ozone levels. There are five months during which every single day has more than 3 hours during which the $O_3$ concentration is relatively low (smaller than or equal to $25\mu g/m^3$), while the remaining 7 months have at least 1/3 of all days with a similar ozone level. We recognize that $NO_2$ and NO conversion processes via $O_3$ do occur, but given that there is a substantial amount of time in which there is insufficient surface $O_3$, changes in $NO_2$ and NO concentrations that do occur must be contributed to more so by emissions, meteorology, transport, other chemical processes, and their thermodynamic conditions at the point of emissions, hence $\alpha_1$.

On top of this point, the geospatial distribution of heavy industrial enterprises in Shanxi are mostly clustered, so they may face similar atmospheric chemical environments and may influence each other, which would lead to those enterprises downwind of the first one having even less ozone available to undergo any titration reactions (Cohen et al., 2011; Cohen and Prinn, 2011). This has been found to be very important, especially so at night time or under heavy aerosol loading conditions when there is insufficient UV to restart such chemical processes, which is commonly observed in the region and consistent with the observations shown.

On top of this, we believe that the data speaks for itself in this case. We have shown the statistics of the distributions of $\alpha_1$ and believe that they are significantly different from each other. We do understand that some ozone titration occurs, but also do not believe it is fair to merely assume that it is important in the context of this work, just because in other places it is dominant, and do not feel it is fair to state that the thermodynamic contribution is not important, when clear statistical differences are observed in the computed PDFs. We also believe that a more careful analysis of the data will show that changes in the $NO_2$ and $O_3$ are not always correlated or anti correlated, with it being clear that there are other processes which occur, including by not limited to transport and changes in aerosol induced UV, as demonstrated in the 2nd Response Figure 2, and as included in other parts of the emissions equations that this work is using.

Thank you for the suggestion to investigate in more detail the issues surrounding upslope winds, plume rise, and differences under different times of the year and different climatological conditions. We believe that this future work will provide interesting insights to better understand at high frequency and spatial resolution the relative importance of all of these driving factors, such as those which are first order, second order, etc.

**2nd Response Table 1**. Ozone concentration statistics from ambient air quality monitoring stations near an iron and steel factory in Taiyuan City of 2019

| Month | Days in which have more than 3 hours that $O_3$ concentration≤5μg/m³ | Days in which have more than 3 hours that $O_3$ concentration≤25μg/m³ |
|---|---|---|
| January | 27 | 31 |

| February | 4 | 20 |
|---|---|---|
| March | 10 | 23 |
| April | 15 | 22 |
| May | 10 | 19 |
| June | 0 | 7 |
| July | 0 | 9 |
| August | 6 | 13 |
| September | 13 | 28 |
| October | 24 | 28 |
| November | 26 | 27 |
| December | 27 | 29 |

[Figure]

**2nd Response Figure 2**. Time series of hourly concentration of $O_3$ and $NO_2$ from ambient air quality monitoring stations near an iron and steel factory in Taiyuan City.

There are many observations in this area that demonstrate at times there is insufficient ozone present at the surface to convert NO to $NO_2$ even on days which have a high surface temperature, which should be the times with the largest surface $O_3$ concentration (Response Figure 2). Statistics of the two figures (2nd Response Table 2) show that more than 56% of the days have at least 3 hours of $O_3$ less than 5ug/m$^3$, and more than 89% if the days have at least 3 hours of $O_3$ less than 20ug/m$^3$. Hence, even in summer there are insufficient ozone time.

**2nd Response Table 2. Ozone concentration statistics for Response Figure 2**

| Day | Hours that $O_3$ concentration $\leq 5\mu g/m^3$ | Hours that $O_3$ concentration $\leq 25\mu g/m^3$ |
|---|---|---|
| Sep. 11th | 0 | 5 |
| Sep. 12th | 3 | 14 |
| Sep. 13th | 10 | 13 |
| Sep. 14th | 4 | 13 |
| Oct. 3rd | 4 | 12 |
| Oct. 4th | 0 | 2 |
| Oct. 5th | 0 | 0 |
| Oct. 6th | 0 | 5 |
| Oct. 7th | 7 | 15 |

2) Response Figure 4: are the PDFs from the six sources really that different? It reads to me like the PDFs of four sources (Cement, Power, Iron&Steel and Alumina) are very similar, and the PDFs of the other two are close to each other. So are the derived alpha1 values really "distinguishing these sites from each other"?

We have calculated the statistical distribution of $\alpha_1$ and have presented the results of the 10th, 25th, 50th, 75th, and 90th percentile values in the 2nd Response Table 3. We hope that the table below makes the differences clearer for the reader to follow. Cement, power and iron/steel have higher $\alpha_1$ at almost all the percentile values than the other three types of sources, with cement factories having the highest values, while boilers clearly have the lowest values. Further details between more closely related sources are also demonstrated. For example, power and iron/steel have significant differences across different parts of the distribution with power having larger values from the 25th through 75th percentiles and steel having higher values at the 90th percentile level. This is consistent with the fact that iron and steel use several different processes, one of which contains very high temperature combustion (blast furnace based) and the other which contains a lower temperature process (sinter bed based). Other such differences are shown within the table and in the plots in the paper.

**2nd Response Table 3.** Distribution of $\alpha_1$ ($10^{th}$, $25^{th}$, $50^{th}$, $75^{th}$, and $90^{th}$ percentile values) calculated based on CEMS using MFIEF at different industry sources.

| Industry Sources | $\alpha_1$ | | | | |
|---|---|---|---|---|---|
| | $10^{th}$ | $25^{th}$ | $50^{th}$ | $75^{th}$ | $90^{th}$ |
| Cement | 1.4 | 2.2 | 3.7 | 6.3 | 10.3 |
| Power | 1.4 | 2.0 | 3.3 | 5.9 | 8.6 |
| Iron/Steel | 1.3 | 1.8 | 3.2 | 5.4 | 9.4 |
| Coke | 1.2 | 1.5 | 2.2 | 3.8 | 6.6 |
| Aluminum oxide | 1.3 | 1.4 | 2.6 | 3.5 | 5.0 |
| Boiler | 1.1 | 1.3 | 1.7 | 2.3 | 3.9 |

I think the other parts of this study is excellent, and I retain the suggestion to greatly reduce the amount of discussion of this part.

Thank you again for helping us to improve how we both interpret and explain the results. We have made updates in the paper accordingly.

**References**

Cohen, J. B. and Prinn, R. G.: Development of a fast, urban chemistry metamodel for inclusion in global models, *Atmos. Chem. Phys.*, **11**, 7629-7656, https://doi.org/10.5194/acp-11-7629-2011, 2011.

Cohen, J. B., Prinn, R. G., and Wang, C.: The impact of detailed urban-scale processing on the composition, distribution, and radiative forcing of anthropogenic aerosols, *Geophys. Res. Lett.*, **38**, https://doi.org/10.1029/2011gl047417, 2011.

Cohen, J. B. and Wang, C.: Estimating global black carbon emissions using a top-down Kalman Filter approach, J. Geophys. Res. Atmos., 119, 307-323, https://doi.org/10.1002/2013jd019912, 2014.

Penning de Vries, M. J. M., Beirle, S., and Wagner, T.: UV Aerosol Indices from SCIAMACHY: introducing the SCattering Index (SCI), *Atmos. Chem. Phys.*, **9**, 9555–9567, https://doi.org/10.5194/acp-9-9555-2009, 2009.

Torres, O., Jethva, H., Ahn, C., Jaross, G., and Loyola, D. G.: TROPOMI aerosol products: evaluation and observations of synoptic-scale carbonaceous aerosol plumes during 2018–2020, *Atmos. Meas. Tech.*, **13**, 6789-6806, https://doi.org/10.5194/amt-13-6789-2020, 2020.